# Harnessing fluorescent carbon quantum dots from natural resource for advancing sweat latent fingerprint recognition with machine learning algorithms for enhanced human identification

**Nisha Yadav**[1], **Deeksha Mudgal**[1], **Amarnath Mishra**[2], **Sacheendra Shukla**[3], **Tabarak Malik**[4]*, **Vivek Mishra**[1]*

1 Amity Institute of Click Chemistry Research and Studies, Amity University Uttar Pradesh, Noida, India, 2 Amity Institute of Forensic Sciences, Amity University Uttar Pradesh, Noida, India, 3 Amity Institute of Applied Sciences, Amity University Uttar Pradesh, Noida, India, 4 Biomedical Sciences, Institute of Health, Jimma University, Jimma, Ethiopia

* vmishra@amity.edu (VM); tabarak.malik@ju.edu.et (TM)

## Abstract

Nowadays, it is fascinating to engineer waste biomass into functional valuable nanomaterials. We investigate the production of hetero-atom doped carbon quantum dots (N-S@MCDs) to address the adaptability constraint in green precursors concerning the contents of the green precursors i.e., *Tagetes erecta* (marigold extract). The successful formation of N-S@MCDs as described has been validated by distinct analytical characterizations. As synthesized N-S@MCDs successfully incorporated on corn-starch powder, providing a nano-carbogenic fingerprint powder composition (N-S@MCDs/corn-starch phosphors). N-S@MCDs imparts astounding color-tunability which enables highly fluorescent fingerprint pattern developed on different non-porous surfaces along with immediate visual enhancement under UV-light, revealing a bright sharp fingerprint, along with long-time preservation of developed fingerprints. The creation and comparison of latent fingerprints (LFPs) are two key research in the recognition and detection of LFPs, respectively. In this work, developed fingerprints are regulated with an artificial intelligence program. The optimum sample has a very high degree of similarity with the standard control, as shown by the program's good matching score (86.94%) for the optimal sample. Hence, our results far outperform the benchmark attained using the conventional method, making the N-S@MCDs/corn-starch phosphors and the digital processing program suitable for use in real-world scenarios.

## Introduction

Driving inspiration from nature to take design cues for the manufacture of functional materials, it is essential to adopt an ecologically benign and renewable raw material source to meet

**Data Availability Statement:** All relevant data are within the paper and its Supporting Information files.

**Funding:** The author(s) received no specific funding for this work.

**Competing interests:** The authors have declared that no competing interests exist.

the significant difficulties confronting the sustainable production of nanomaterials without utilizing hazardous elements. Carbon dots (CQDs) are a greener example of a functionalized material formed by digesting biomass with a specific natural composition and intriguing properties [1,2]. The field of naturally occurring CQDs has grown considerably; nevertheless, owing to poor characterization of eco-friendly precursors, which are flexible and diverse in nature, the synthesis approach and repeatability remain a bottleneck. The use of edible resources for the synthesis of CQDs, such as tomato, banana, milk, orange, cabbage, maize bran, and coconut, has lately received a lot of attention [3]. However, using non-edible green materials would have obvious benefits and accurate green precursor characterization which would provide a better understanding of the synthesis mechanism and help with reproducibility [4]. We explored the optical properties of Tagetes erecta (Marigold extract) derived hetero-atom doped carbon quantum dots (N-S@MCDs) [5]; hence it is intriguing to investigate its precursors and intermediates' behavior. It's vital to note that the initial source of carbon has a significant impact on the characteristics of carbon quantum dots. Chemically synthesized CQDs have poor yields, hazardous precursors, and a need for separate passivation, which has an impact on capital costs and the environment [2]. On the other hand, biomass-derived CQDs are advantageous because of their potential to be produced on a wide scale, sustainability, availability, renewability, green approach, and self-passivation. Owing to its inherently complex composition of organic molecules, diverse functional groups, and hetero-atom dopants, CQDs produced from different sources of biomass exhibit a wide range of characteristics. A typical example is the co-doping of nitrogen, sulfur, and phosphorous in CQDs by using onion peels because these materials already contain polyphenols and minerals [6]. However, using pulp-free lemon juice produces red-emitting CQDs, demonstrating the impact of the biomass precursor on the optical characteristics. The majority of CQDs formed from biomass are blue-emitting, revealing a common optical characteristic observed in this class of carbon nanomaterials made from naturally occurring substances [7].

Fingerprint analysis has been recognized as a trustworthy method for human identification since the 19th century. When a finger contacts a solid object, its secretion leaves a distinctive ridge pattern on the surface. Because of their poor optical contrast with the substrates, these patterns are frequently referred to as latent fingerprints (LFPs), which are invisible to the human eye [8,9]. Two primary studies should be carried out very carefully and patiently to detect and identify LFPs. One is the creation of LFPs, which often requires brushing different dyes to enhance the disparity between fingerprints and backgrounds before imaging techniques are used to record the patterns. The alternative method is LFPs comparison, which involves gathering highly comparable patterns from many sources, comparing these patterns using common software, and then reliably identifying the patterns based on the matching score [8,10–13]. Numerous techniques have been created so far to find LFPs [14–17]. Fluorescence imaging is one of them, and it's a common technique where LFPs are coated with dyes or phosphors nanomaterials to provide elevated fluorescent signals when illuminated [18]. For the development of LFPs, for instance, magnetic particles [19], CQDs [20], and up-conversion nanoparticles (UCNPs) [21] have been used. However, several dye powders, especially rhodamines, tend to stick to substrates' overall surface, making it difficult to discriminate between fingerprint lines. In practice, these dyes are dispersed in a substantial amount of medium (powder carrier) to provide adequate mobility while avoiding dye powder stagnation and substrate adherence. We also discovered a spraying approach that produces LFPs on impenetrable surfaces using CQDs based on the coffee ring effect and unquenched CQDs fluorescence throughout the drying process [22]. These techniques' limitations, however, frequently decrease from their time-consuming or expensive preparation processes, potential toxicity, destructive detection procedures, and poor stability in real-world environments [23]. As a

result, creating a novel, high-quality material that can easily, safely, and portably light up LFPs on multiple substrates while also creating a quantitatively analytical program that can compare LFPs with accuracy and ease has become a pressing issue for LFP detection and identification.

In this study, we have engineered a novel nanocomposite fluorescent fingerprint powder formulation (N-S@MCDs/corn-starch phosphors) for the visualization of LFPs. This formulation consists of environmentally friendly, highly fluorescent N-S@MCDs synthesized from marigold extract. When integrated into corn starch powder through an eco-friendly process, these N-S@MCDs enable the development of exceptionally fluorescent fingerprint patterns on various non-porous surfaces. These patterns can be rapidly enhanced under UV light, resulting in the presentation of vivid, well-defined fingerprints that can be preserved over an extended period. We used a digital-processing tool to accurately detect LFPs, which retrieved the fine characteristics of LFPs on various substrates, a precise comparison of the characteristics with the control, and then displayed the corresponding scores on a computer. The highest possible matched score was 86.94%. Additionally, we have elucidated the conversion mechanism by which marigold extract is transformed into N-S@MCDs, shedding light on the intermediate stages of this process. The translation of these promising research outcomes into practical applications with significant societal and economic implications calls for a collaborative and interdisciplinary effort, involving researchers from the fields of materials science, biology, and forensic investigation.

## Experimental section

### Materials

Flower waste from temples was assessed for the synthesis of blue-emitting, environment-friendly highly fluorescent carbon quantum dots. Ethylenediamine (EDA), methionine, corn starch, Coomassie blue (dye) was obtained from Sigma-Aldrich and utilized without additional purification. SD Fine Chem supplied the solvents utilized in the synthesis, 100% ethanol and glacial acetic acid. All investigations and photo-physical studies were conducted using Milli-Q water. All characterization studies were conducted at room temperature.

### Synthetic procedure

**Synthesis of hetero-atom doped carbon quantum dots (N-S@MCDs).** Hetero-atom doped carbon quantum dots (N-S@MCDs), which serve as nano-carbogenic highly fluorescent powder carriers, were produced using a mild hydrothermal method. The petal of the marigold flower was initially subjected to crushing using a domestic mixture, followed by filtration to obtain the marigold extract (ME). One frequently used methodology comprises combining a mixture of 10 mL of ME with 300 μL of ethylenediamine and 300 mg of methionine. In order to produce a homogenous solution (ME-1), the liquid was agitated for a duration of 10 minutes. Subsequently, the solution was introduced into an autoclave equipped with a Teflon lining and subjected to hydrothermal treatment for a duration of 5 hours at a temperature of 180˚C. The product suspension underwent centrifugation in order to eliminate sizable particles, following the cooling of the autoclave to ambient temperature. The acquired sticky black product was lyophilized, and the resulting supernatant (N-S@MCDs) was properly re-dispersed in water [5]. Moreover, to clarify the reaction mechanism, four different reactions under comparable reaction environments were carried out with reaction time varying from 1h to 4h. The resulting samples were known as ME-1h, ME-2h, ME-3h, and ME-4h respectively.

**Preparation of N-S@MCDs/corn-starch phosphors.** N-S@MCDs and corn starch are combined in a mortar, and then Milli Q water is added as a binder, followed by ultrasonic agitation for 10 min at room temperature [24,25]. The resultant reaction mixture was air-dried,

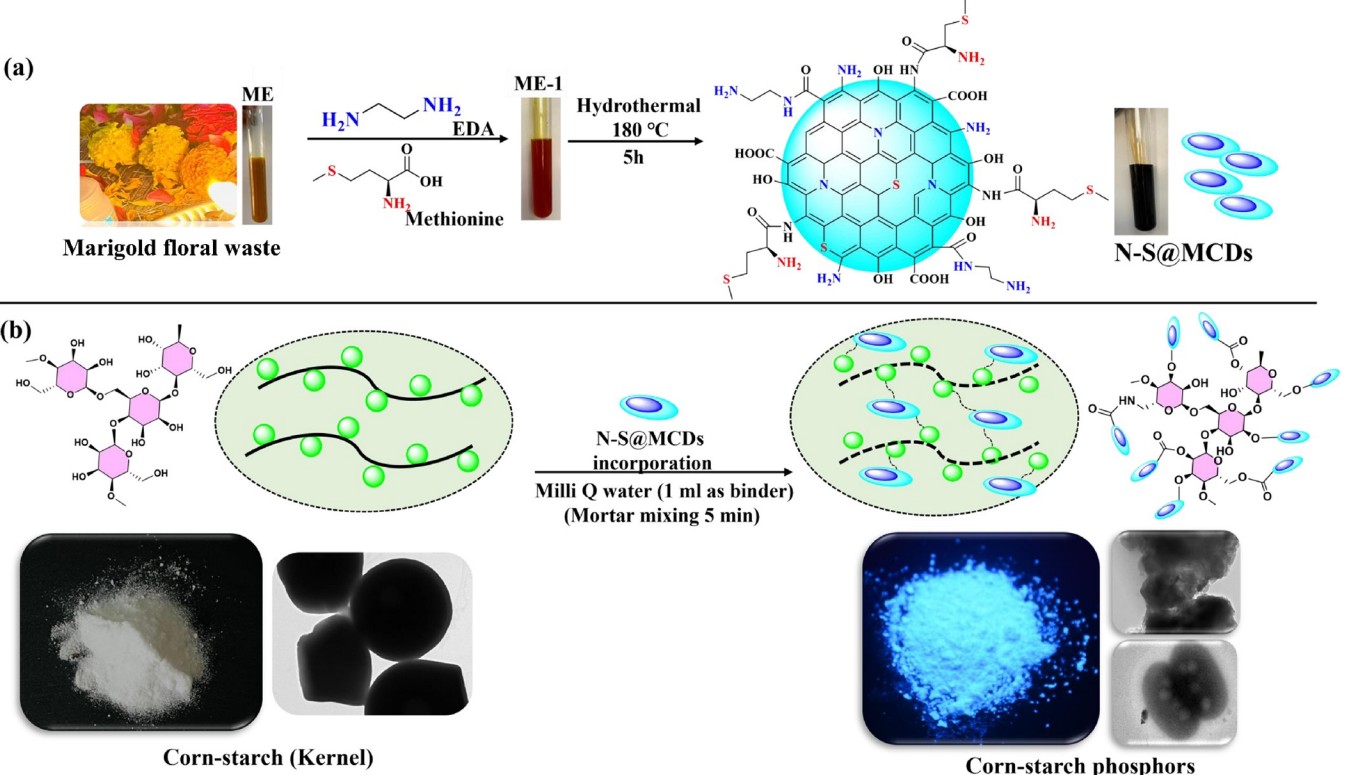

**Fig 1.** (a) Schematic illustration of synthesis of N-S@MCDs from temple floral waste, (b) Incorporation of N-S@MCDs onto the surface of corn-starch.

and a pale-yellow powder, luminous corn starch was produced as a fingerprint-developing powder composition (as illustrated in Fig 1).

## Characterization and Instrumentations

**Morphology.** The high-resolution X-ray diffraction (XRD) pattern was produced using a Cu Kα radiation source and a PANalytical Empyrean X-ray diffractometer at an accelerating potential of 40 kV, in the 2θ range of 10–60˚. Finely powdered solid samples had been acquired for X-rays analysis. Transmission electron microscopy (TEM) and high-resolution transmission electron microscopy were used to assess the surface morphology and particle size of N-S@MCDs (HRTEM). TEM images were obtained on a JEOL JEM-2100F electron microscope with a 200 kV acceleration voltage. The material was drop-cast onto a copper grid that had been coated with carbon, and then the samples were allowed to dry at room temperature. Scanning electron microscope (SEM) with EDAX (Energy-dispersive X-ray) spectroscopy was done from the University of Delhi using JEOL JSM 6610LV using Tungsten electron sources at 10KV with magnification X5 to X 3,00,000 with high vacuum mode using LN2 free EDS detector. $^1$H and $^{13}$C NMR (400 MHz for proton and 100 MHz for carbon) spectra were recorded with a JEOL spectrometer using trimethylsilane as an internal standard.

**Chemical composition.** Fourier transform infrared spectra (FTIR) were obtained using a Nicolet-5DX FTIR spectrophotometer, covering the spectral range of 400–4000 cm$^{-1}$. Furthermore, the elemental composition was analyzed using Raman spectroscopy and X-ray photoelectron spectroscopy. The X-ray photoelectron spectroscopy (XPS) data were obtained using a PHI Versa Probe II instrument equipped with Al Kα radiation. A LabRam HR spectrometer

with backscattering geometry, a CCD-detector, and a 785 nm Ar laser were utilized to perform Raman spectroscopy at a 50× magnification.

**Optical characterization of <u>N-S@MCDs</u>.** The optical and photoluminescence (PL) measurements for N-S@MCDs and N-S@MCDs-coated corn-starch were monitored through Duetta Absorbance and Fluorescence Spectrometer in the wavelength range 200–800 nm using a xenon lamp as an excitation source.

**Visualization of LFPs using N-S@MCDs/corn-starch phosphors; powder composition.** The fingerprints were generated using the usual powder brushing approach, employing N-S@MCDs/corn-starch phosphors as a labeling marker. LFPs were collected from many donors, utilizing a range of filtering and non-filtering surfaces, including mono background color surfaces (such as glass slides, paper, and metallic surfaces) as well as intense background auto-fluorescence surfaces (such as cash). All the fingerprints included in this study were obtained exclusively from the thumb of a female participant aged 25 years. Prior to obtaining the fingerprints, the donor's hand underwent a meticulous cleansing process using soap and water, followed by air drying. Subsequently, the sterile fingertips gently massaged the foreheads of the contributors, exerting pressure on diverse textures. Latent fingerprints were captured utilizing a Canon EOS 200D II 24.1MP DSLR camera, under the illumination of a 365 nm ultraviolet light source.

In adherence to the regulations set forth by the Federal Bureau of Investigation (FBI) [26], a control sample was prepared by rolling the same figure onto the black inking plate. This was done by applying ink from the crease of the first joint to the tip of the finger, as well as from one edge of the nail to the opposite edge. To mitigate the occurrence of smearing, it is advised to delicately elevate the finger. The photographs were subsequently taken during daylight hours utilizing the identical camera., to prepare a control, the same figure was rolled immediately on the black inking plate to create a control by coating it with ink from the first joint's crease to the tip of the finger, as well as from one edge of the nail to the other. In order to prevent smudging, carefully lift the finger. The pictures were afterward captured in daylight using the same camera [13,27].

## Benchmark

The classical dye-soaked cotton pads as an "ensemble material" advances a new strategy for the development of latent fingerprints [28]. Typically, 0.02 g of Coomassie blue was dissolved in a solution containing 1 mL of glacial acetic acid and 9 mL of absolute ethanol. Then, a little cotton pad was first dipped into the dye solution. The cotton pad was used to gently apply the dye solution onto the LFP substrate until it covered the fingerprint. The extra dye solution was then removed from the fingerprint by rinsing it with the rinse solution. The produced fingerprint was then allowed to dry naturally until all of the rinse solutions on the substrate had evaporated.

## Results and discussion

### Synthesis of nano-carbogenic powder composition (N-S@MCDs incorporated corn-starch)

In this study, a straightforward method was developed to produce a sustainable nano-carbogenic powder carrier known as hetero-atom doped carbon quantum dots (N-S@MCDs). This environmentally friendly material was synthesized using low-cost marigold extract as a precursor. The utilization of marigold extract as a green carbon source has facilitated the achievement of sustainability goals by circumventing laborious and intricate procedures that involve the handling of hazardous substances and the fabrication of templates. This approach also

contributes to the reduction of waste generation, aligning with the principles of zero waste. Furthermore, the utilization of non-consumable green substances would confer benefits, and precise evaluation of green precursors would enhance replicability and foster a more comprehensive comprehension of the synthesis mechanism. In this study, we additionally investigated the optical characteristics of carbon quantum dots generated from Tagetes erecta, often known as marigold extract. Consequently, it is of great interest to examine the precursors and intermediates involved in the synthesis of these carbon quantum dots. The N-S@MCDs were synthesized using the green hydrothermal method, which yielded very stable and conductive properties when immersed in an aqueous environment. Even after six months, no precipitation was seen. Furthermore, four separate reactions were carried out under identical conditions, with reaction periods spanning from 1 to 4 h, to better understand the mechanistic process of synthesized N-S@MCDs. As a result, samples ME-1h, ME-2h, ME-3h, and ME-4h were produced. Additionally, based on their diverse structure and chemical composition, CQDs incorporated on the biomass-based greener surface (corn starch powder) to develop a fingerprint powder composition i.e., N-S@MCDs/corn-starch phosphors, by following a greener method within 5–10 min of grinding in mortar by using milli-Q water as a binder, allowing for rapid visual enhancement for all the surfaces, even the best visibility is observed on plastics and glasses under UV light along with long-time preservation of developed latent-fingerprint on distinct surfaces.

The intrinsic structural and morphological characteristics of hetero-atom doped carbon quantum dots (N-S@CDs) and nano-carbogenic powder composition (N-S@MCDs/corn-starch phosphors) were revealed by using a variety of characterization techniques, including powder x-ray diffraction (PXRD), high-resolution transmission electron microscopy (HRTEM), scanning electron microscopy (SEM), Fourier transform infrared (FTIR) spectroscopy, X-ray photoelectron spectroscopy (XPS), Raman, and photoluminescence (PL) analysis.

**Morphological characterization.**   PXRD analysis was utilized first to determine the morphological and structural properties of as-synthesized N-S@MCDs and N-S@MCDs-coated corn starch. The PXRD-pattern of N-S@MCDs shown in Fig 2 shows a broad (002) diffraction peak centered about 25.4˚ (2), which corresponds to an interlayer spacing of 3.5, indicating the amorphous nature of the material. The related d-spacing of N-S@MCDs is greater than that of bulk graphite, at 3.34, indicating a turbostratic carbon structure (sp2 deformation). A further peak, centered around 42.50˚ (2), represents a mainly graphitic structure with an interlayer spacing of 0.21 nm [29].

Furthermore, TEM and HR-TEM images were used to investigate the particle form, size, and nature of N-S@MCD. Fig 3A–3F shows that as-prepared N-S@MCDs were uniform, and S1 Fig (see supporting information) confirmed the distribution of mono-dispersed spheres with a mean diameter of 6.4±1.64 nm. The HR-TEM image confirms the structural integrity of the N-S@MCD and reveals the presence of particles with clearly defined lattice fringes, characterized by a d-spacing of 0.35 nm. The integration of N-S@MCDs onto the surface of corn starch, derived from corn kernels, resulted in oval-shaped particles with smooth surfaces. These particles ruptured because of hydroxyl bonding between the synthesized N-S@MCDs and the surface and voids of the corn starch. This observation is further supported by the SEM images obtained (Fig 3H and 3I).

Hereby, the obtained nano-carbogenic powder composition (N-S@MCDs/corn-starch phosphors) with particle size around 0.5–5 µm appear agglomerated due to the presence of hydroxyl bonding; exhibit uniform blue fluorescence indicating homogenous incorporation of N-S@MCDs on corn-starch. SEM image was used to further confirm the morphology of N-S@MCDs and N-S@MCDs-coated corn starch, as depicted in Fig 3G–3I. Herein, SEM morphology revealed that the graphitic rod is removed during the carbonization process, resulting

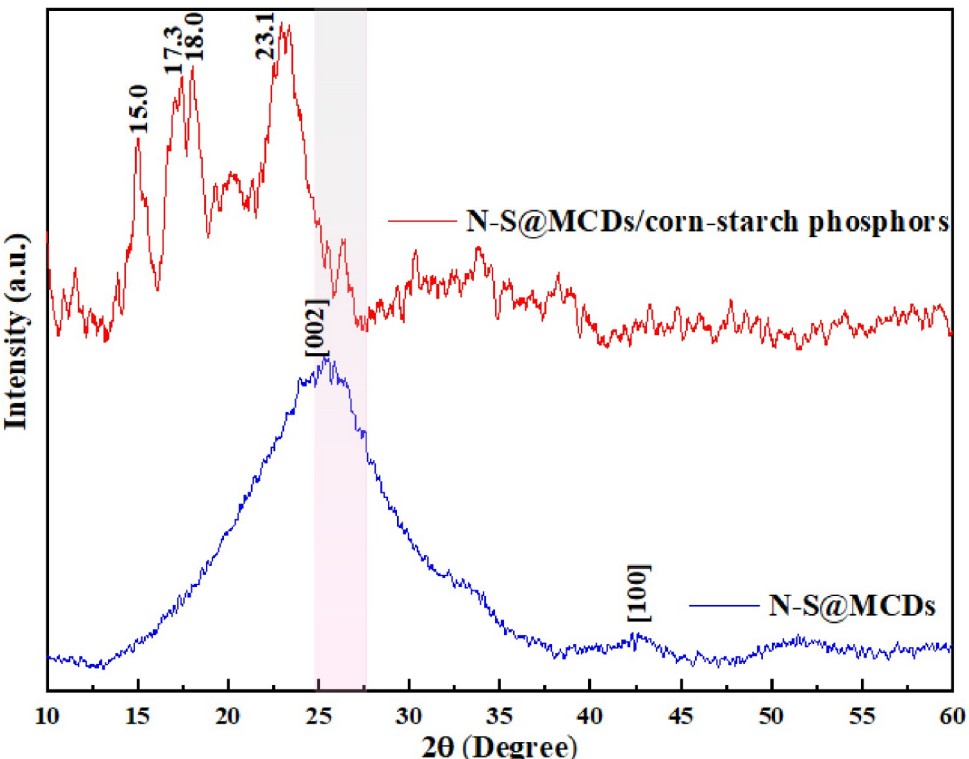

**Fig 2. PXRD pattern of N-S@MCDs and nano-carbogenic fingerprint powder composition (N-S@MCDs/corn-starch phosphors).**

N-S@MCDs that are approximately spherical in shape, uniform in size and distribution, and with a slight agglomeration effect. Meanwhile, due to the presence of N-S@MCDs on the surface of corn starch, the nano-carbogenic fingerprint powder composition exhibit a ruptured oval shape which is also confirmed by the TEM images as shown in Fig 3C–3F. Hence, the morphological changes are in good agreement with the outcome achieved by XRD, SEM, TEM analysis, confirming the successful incorporation of N-S@MCDs onto the surface of corn starch.

**Spectral analysis (FTIR, Raman and XPS).** In this context, FTIR spectra were recorded in the presence of various surface functionalities to investigate the successful synthesis of N-S@MCDs from the marigold extract. As depicted in Fig 4, the absorption band at around 3476, 2960, 2860, and 1695 $cm^{-1}$, by the stretching vibrations of the O-H. N-H, C-H ($sp^3$), and C = C functionalities, respectively [30]. Additionally, similar adsorption bands with intensities ranging from 800 to 1400 $cm^{-1}$ were found in ME, ME-1, and N-S@MCDs and were ascribed as stretching vibrations of the C-C, C-O, C-S, and C-H groups, respectively. Nevertheless, the existence of C-O groups was verified by the presence of the strong band at 1231 $cm^{-1}$, which was identified solely in the ME sample [31]. Intriguingly, the C = O groups' stretching vibration band was identified at 1773 $cm^{-1}$ in the N-S@RCD sample. Moreover, the C = N and C-N bands are exclusively visible in N-S@RCD, as evidenced by the stretching bands at 1695 and 1387 $cm^{-1}$ [32]. The FTIR spectra of N-S@MCDs and N-S@MCDs-coated corn-starch both show similar stretching vibration, which is an indication that N-S@MCDs is coated onto and in voids of corn-starch particles. As a result, the adsorption effects between N-S@MCDs and the fingerprint residues, which are based on the functional groups of N-S@MCDs indicated above [33–35], will function at their full potential. Additionally, the degree of structural disorder in the carbon matrix was evaluated subjectively and quantitatively using a quasi-technique, i.e., Raman spectroscopy.

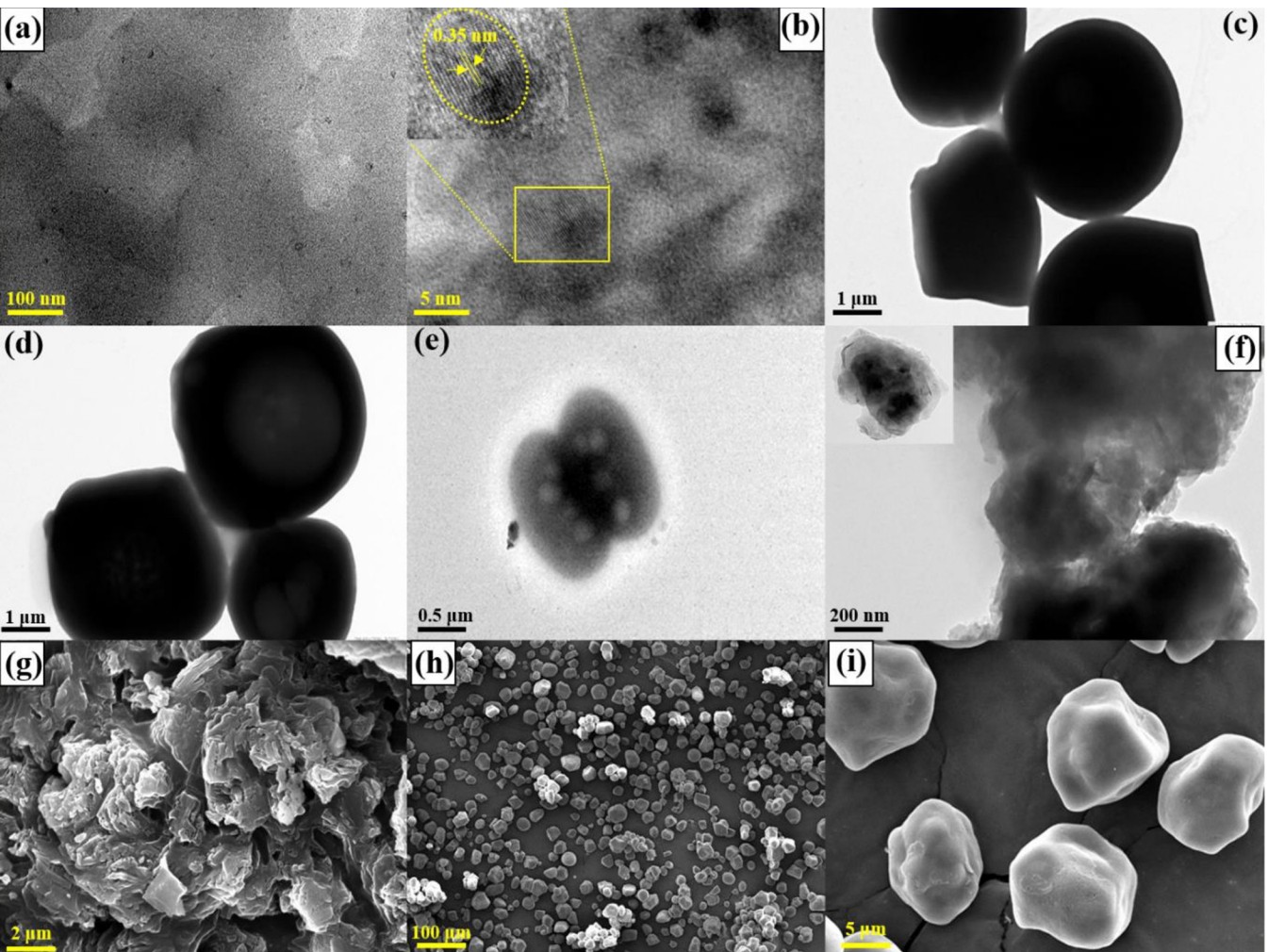

**Fig 3.** TEM and HR-TEM images of N-S@MCDs (a, b); corn-starch (c, d); N-S@MCDs/corn-starch phosphors (e, f (inset: TEM image at 100 nm)); SEM images of N-S@MCDs (g); N-S@MCDs/corn-starch phosphors (h, i).

Accordingly, the Raman spectroscopy of N-S@MCDs and N-S@MCDs-coated corn-starch was recorded. As shown in Fig 5, the Raman spectrum of N-S@MCDs, two predominantly peaks observed at 1350 cm$^{-1}$ and 1546 cm$^{-1}$, which are frequently attributed, respectively, to the disordered D- and crystalline G-bands [36]. The D-band to G-band intensity ratio ($I_D/I_G$) of the N-S@MCDs was roughly 0.87, showing the presence of a significant proportion of amorphous carbon in the synthesized N-S@MCDs-structural composition. Moreover, the Raman signals of corn-starch show majority of bands due to coupled hydrogen vibrations; for example, with respect to CH, CH$_2$, and C-OH deformations, one band can be seen at 1,461 cm$^{-1}$; while the CCH and COH deformation modes are coupled in the range between 1,380 and 1,400 cm$^{-1}$. However, bands with contributions from many vibrational modes including CO and CC stretching as well as COH and CCH deformations, are visible in the region between 1,340 and 1,200 cm$^{-1}$. This led to the conclusion that the region between 1,500–1,200 cm$^{-1}$ contains a rich of structural information, and that the Raman spectra of corn starch exhibit a variety of vibrational characteristics in this range. Meanwhile, in the same region, N-S@MCDs-coated corn-starch shows a broad vibrational band, which corelated to the coupled vibrations

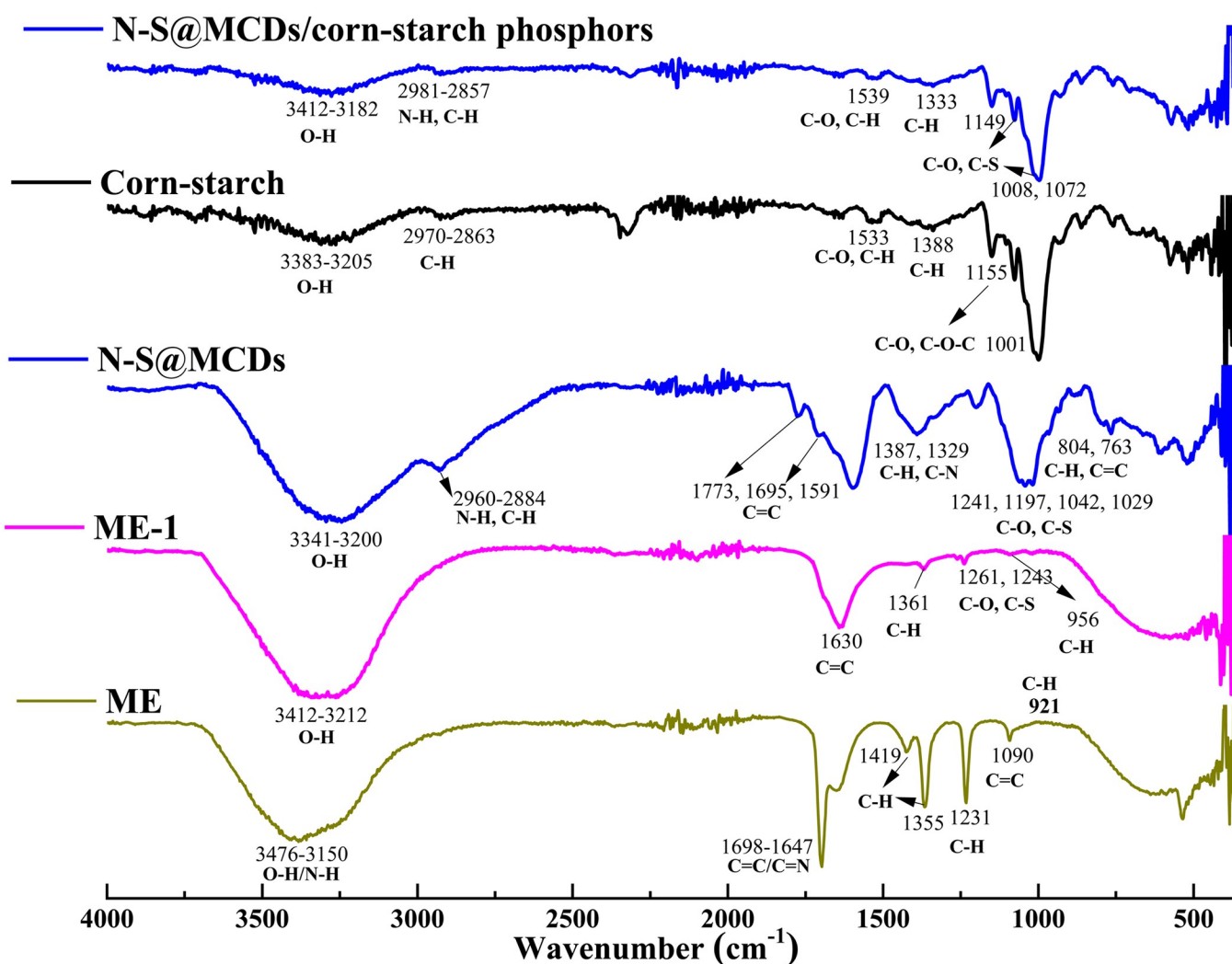

**Fig 4. FT-IR spectra of marigold extract (ME), ME-1, N-S@MCDs, corn-starch and N-S@MCDs/corn-starch phosphors.**

involving hydrogen atoms i.e., CH, $CH_2$, COH, CO and CC-deformations during incorporation process [37]. Therefore, the emergence of a broad signal in N-S@MCDs served as evidence for the successful incorporation of N-S@MCDs into the surface and in corn-starch voids.

Hereafter, to provide more substantial proof of the elemental composition and surface states of N-S@MCDs and N-S@MCDs/corn-starch phosphors, X-ray photoelectron spectroscopy (XPS) was conducted. The presence of C1s, N1s and O1s electrons, which are essential components of N-S@CQDs, is responsible for the dominating binding energy peaks at 285.06, 398.40, and 531.35 eV. However, an additional peak at 167.84 eV was seen in N-S@MCDs and is attributed to the S2p electrons (as depicted in Fig 6A). Moreover, the high-resolution C1s spectrum in Fig 6B, features three peaks at 284.49, 284.28, 285.83 and 287.55 eV, that were credited to $sp^2$-graphitic structure (C-C/C = C), C-N, C-S/C-O (epoxy and alkoxy), and C = O species, respectively [31]. The N1s spectra showed two prominent peaks (Fig 6D), indicating the existence of both pyridinic N (398.96 eV) and N-H (401.15 eV), which was consistent with the FTIR data. The presence of C-O/C = O is also confirmed by the deconvolution of the

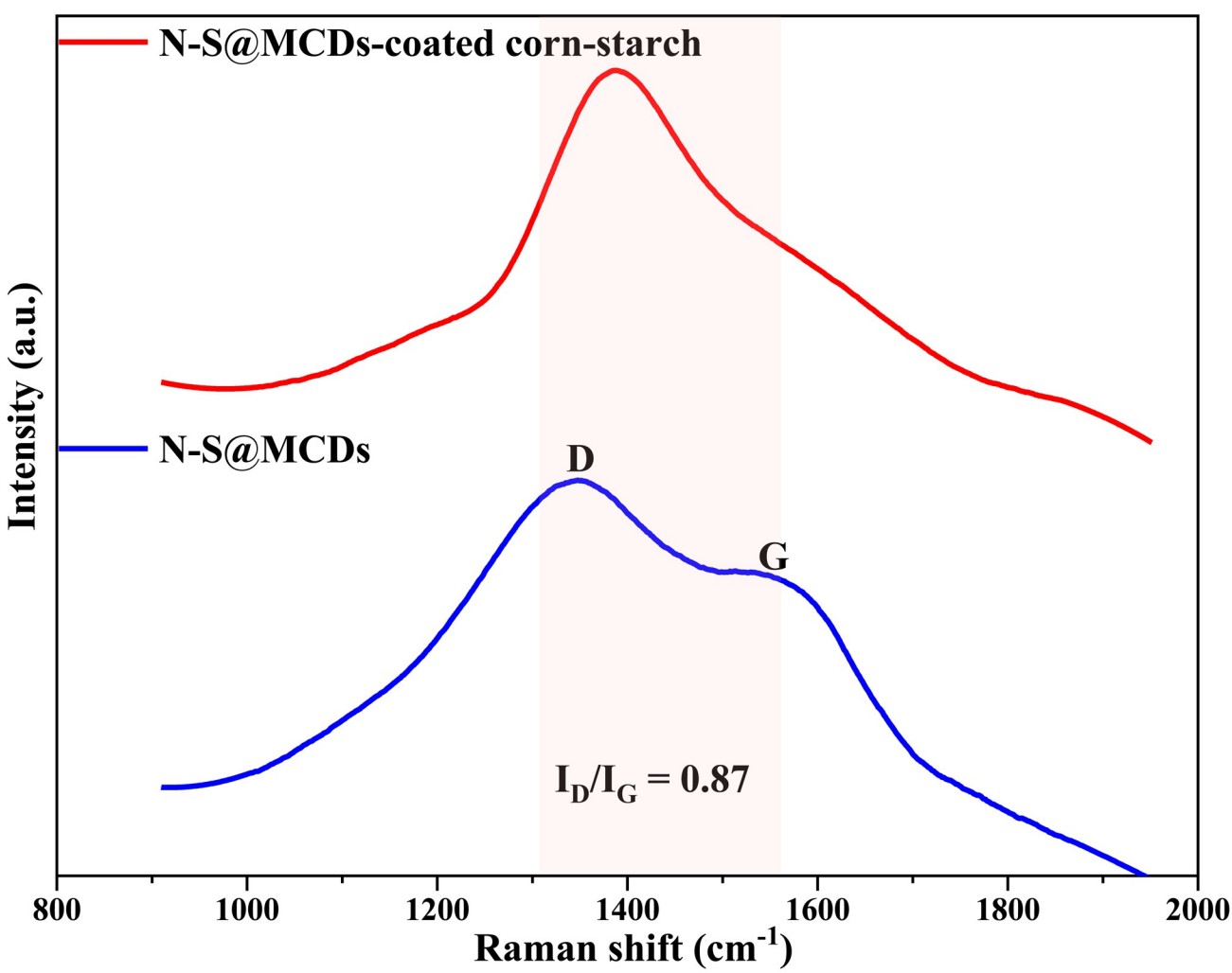

**Fig 5. Raman spectra of N-S@MCDs and N-S@MCDs/corn-starch phosphors.**

oxygen signal at 530.60 eV in Fig 4(E), whereas 532.35 eV are derived from isolated hydroxyl (O-H) (Fig 6C). The S2p spectra, as shown in Fig 6D, principally consisted of three peaks with respective centers at 167.86, 168.85, and 167.23 eV. Due to their spin-orbit interactions, the first two peaks might be assigned to $2p_{3/2}$ and $2p_{1/2}$ of the C-S covalent-bond [38]. Hereafter, as shown in Fig 6A, the survey scan of N-S@MCDs/corn-starch phosphors reveals the composition of carbon, nitrogen, oxygen, and sulphur as C1s (78.17%), N1s (4.62%), and O1s (16.47%), and S2p (0.75%) respectively. This result confirms the successful capping of N-S@MCDs onto corn-starch surfaces is what causes the increased percentage composition of C and O.

Fig 6F displays the deconvolution of the C1s peak. In Fig 6F, the primary peak (C-C/C = C) at 285.15 eV corresponds to $sp^2$-graphitic structure, whereas the peaks at 284.30 and 286.17 eV are assigned to C-N and C-O/C-S/C-O-O, respectively. For more confirmation the presence of the two prominent deconvolution peaks of N1s (398.74, 400.37 eV) and three S2p (167.27, 167.78, and 168.77 eV) signals clearly indicates the successful capping of N-S@MCDs onto corn-starch (Fig 6H and 6I). Additionally, presence of one more peak at 288.06 eV, corresponds to O-C = O in the deconvolution of C1s as well as the presence of three principal peaks

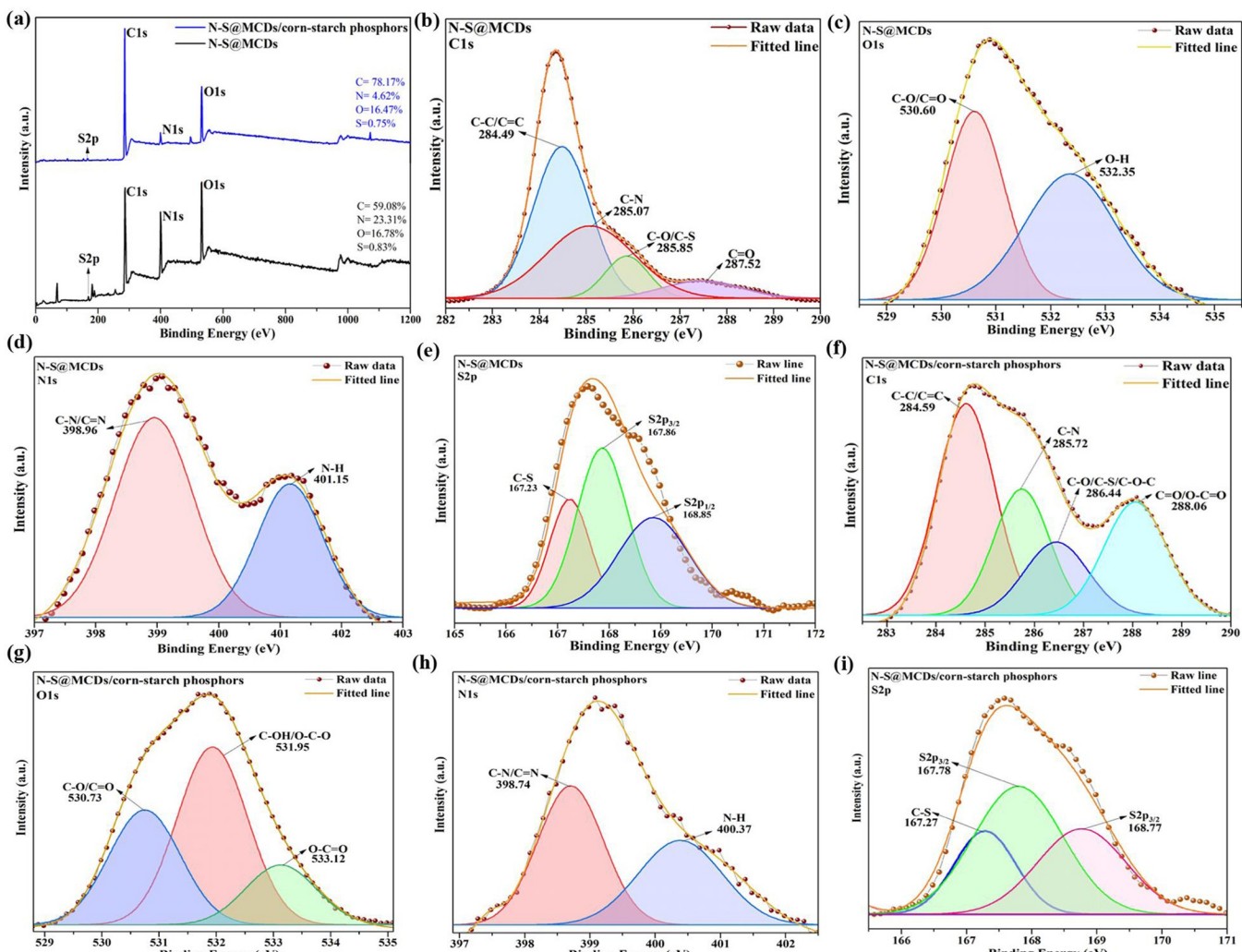

**Fig 6.** XPS spectrum of N-S@MCDs and N-S@MCDs/corn-starch phosphors; (a)survey spectrum; high-resolution spectrum of N-S@MCDs, C1s (b); O1s (c); N1s (d); S2p (e), High-resolution C1s (f); O1s (g); N1s (h); S2p (i) spectrum of N-S@MCDs/corn-starch phosphors.

with respective centers at 530.73, 531.95, and 533.12 eV correspond to C-O/C = O, C-OH/O-C-O, and O-C = O respectively in the deconvolution of O1s (as depicted in Fig 6F and 6G), clearly indicate the successful capping of N-S@MCDs onto corn-starch surface. Furthermore, the presence of carbon (C), nitrogen (N), oxygen (O), and sulfur (S) in N-S@MCDs and N-S@MCDs/corn-starch phosphors is verified by the EDAX spectrum (see S2 and S3 Figs in the supporting information). The elemental composition of N-S@MCDs and N-S@MCDs-coated corn-starch was analyzed, and the findings are presented in S1 Table, which may be found in the supplementary information. Interestingly after incorporation of N-S@MCDs onto the surface of corn-starch, the C/N and C/S ratio increased, which indicates the successful homogenous incorporation of N-S@MCDs. This might be because improved carbonization causes the carbon content of N-S@MCDs to increase as reaction temperature rises. Dehydration, on the other hand, lowers the oxygen concentration of N-S@MCDs.

**Optical behaviour of N-S@MCDs and N-S@MCDs/corn-starch phosphors.** In order to assess the optical characteristics of N-S@MCDs and N-S@MCDs/corn-starch phosphors, UV-

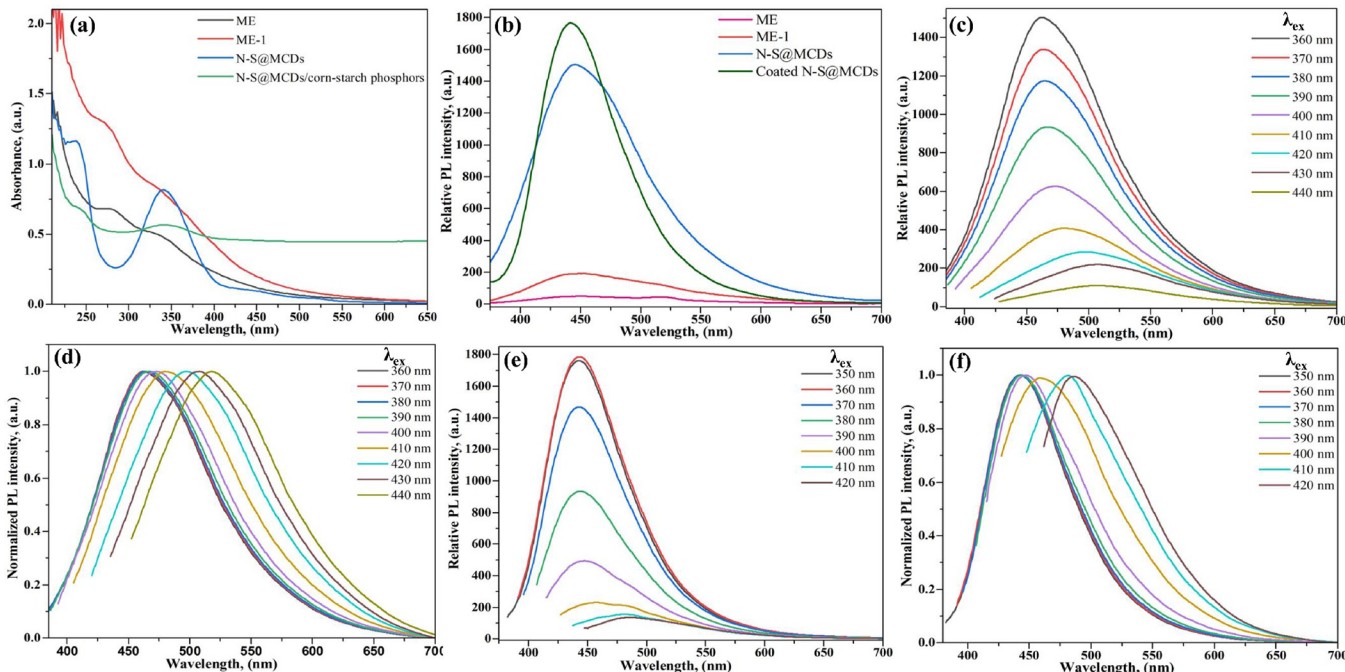

**Fig 7.** Optical behaviour of synthesized N-S@MCDs and N-S@MCDs/corn-starch phosphors; (a) UV-vis absorption spectra; (b) fluorescence spectra; wavelength tuned emission and normalized spectra of N-S@MCDs(c) & (d); and N-S@MCDs/corn-starch phosphors (e) and (f).

visible and photoluminescence (PL) analyses were conducted, as depicted in Fig 7. In comparison to ME-1, the UV-visible spectra of ME displays distinct peaks at 215 nm, 279 nm, and 336 nm, as depicted in Fig 7A. Unlike ME-1, which showcases wide edges at 275 nm and 330 nm. Furthermore, the UV-visible spectra of the N-S@MCDs exhibit two distinct absorption peaks, located at around 240 nm and 345 nm. These peaks correspond to the $\Pi$-$\Pi^*$ and n-$\Pi^*$ transitions, respectively. The N-S@MCDs possess remarkable water-dispersibility and have a pale-yellow hue when observed under normal lighting conditions. Furthermore, upon exposure to UV radiation using a trans-illuminator with a wavelength of 375 nm, these N-S@MCDs exhibit a notable degree of blue fluorescence. Furthermore, it is worth noting that N-S@MCDs/corn-starch phosphors demonstrate a significant blue shift in comparison to N-S@MCDs, with shifts occurring at around 230 and 340 nm. This intriguing phenomenon warrants further discussion. When nanoparticles combine onto the surface of a solid powder, a near-by overlap occurred between the molecular orbitals of N-S@MCDs. The average band gaps of N-S@MCDs were reduced because of the molecular orbital overlap, which caused a red shift in the absorption curve. The above-mentioned red shift again minimized, and the absorption curve shows a blue shift after dispersion of N-S@MCDs into 40 times of starch, since the average distance between CQDs are greatly increased. Conversely, the surface states of N-S@MCDs have a major role in the absorption of those molecules in the visible spectrum. When the molecular orbitals of the surface groups of N-S@MCDs couple with those of corn-starch, the electrons of N-S@MCDs will have higher excited state levels and can absorb light with a wider spectrum. Therefore, substantial interactions between N-S@MCDs and corn-starch are indicated by both the peak blue-shifting and the broadening of the absorption spectra [25,39]. Moreover, the photoluminescence spectra of N-S@MCDs reveals emission maxima at 450 nm for the excitation at 350 nm, while ME and ME-1 exhibits faint fluorescence. Even though starch doesn't have fluorescence, N-S@MCDs/corn-starch phosphors inherit this

character from N-S@MCDs (as shown in Fig 7B). Additionally, the N-S@RCD and N-S@MCDs/corn-starch phosphors displayed a tailored excitation wavelength, maximum emission, and intensity of emission, which is a prominent aspect of many reported CQDs. As the excitation wavelength is increased, N-S@RCD and N-S@MCDs/corn-starch phosphors both exhibit a red shift in the emission wavelength and a quenching of fluorescence intensity.

The luminous starch powder was depicted in S4 Fig (see supporting information). When exposed to UV light, the powder gave off a blue fluorescence with 365 nm excitation, which agrees with fluorescence spectra. Nearly, all CQDs have solution state photoluminescence since they needed to mix with water's hydroxyl group to emit photoluminescence [40]. There are very few reports on the fluorescence of carbon dots in the solid state as it fades along with the loss of water's hydroxyl group if they are in the dry state. On the other hand, coating biological materials with CQDs, which consist of a significant amount of hydroxyl, carboxyl, and amidogen, and enhancing fluorescence properties and even emitting fluorescence in the solid state (S8 and S9 Figs.; see the supporting information). Starch is a biomaterial that contains a lot of hydroxyl groups, when N-S@MCDs is coated on their surfaces, it exhibits increased fluorescence emissions much like it does in solution. Additionally, using quinine sulphate as a reference, the quantum yield of N-S@MCDs was discovered to be 11.78%, which is reasonable when compared to other previously reported green synthesized CQDs (S2 Table).

## Mechanism insight of synthesis of N-S@MCDs from marigold extract

Nuclear magnetic resonance (NMR) and electron spin ionization-mass spectroscopy (ESI-MS) were used to further understand the mechanism of N-S@MCD production from marigold extract. The LC-MS chromatogram was taken after dissolving 1.0 mg of the ME, ME-1, and N-S@MCDs samples in 5.0 mL of $CH_3OH$ at RT (25°C). Low molecular weights of 331.15, 439.19, and 441.45 g mol$^{-1}$ can be seen in the LC-MS chromatograms of the ME and ME-1 samples, which are shown in S7 Fig (see the supporting information). These molecular weights may be connected to various cellulosic wastes that have been identified in marigold extract, respectively. Furthermore, given that N-S@MCDs from the hydrothermal treatment of ME-1 have a relatively higher molecular weight, 679.65 g mol$^{-1}$, it is possible that polycondensation of low molecular weight moieties, like furfural (96 g mol$^{-1}$) and 5-hydroxymethyl furfural (5-HMF), which results in a high molecular weight of N-S@MCDs, occurred during the hydrothermal treatment. Numerous studies have conclusively shown that hydrothermal treatment affects molecular polycondensation [41].

The $^1$H-NMR spectrum of ME, ME-1 and intermediates generated at distinct time interval [1-4h] are shown in Fig 8A, 8C, 8G, 8I and 8K). The peak detected in the range δ; 0.5–1.79 ppm in the $^1$H-NMR spectra of ME, ME-1 and ME-1h/2h/3h/4h are shown in Fig 8, to identify the presence of sp$^3$/sp$^2$/sp alkyl groups. Aside from that, all samples showed the typical peaks associated with alkyl chains connected to -C = O groups, such as aldehyde/esters/ketone (alkyl-/aryl-CHO; alkyl-/aryl-$CH_2$-COO-alkyl/aryl, aryl/alkyl-CO- alkyl/aryl) or allylic protons (Ar- $CH_3$), in the range δ = 2.0–3.0 ppm. This analysis demonstrates that there is a possibility that aryl rings are coupled with alkyl/aryl moieties across ≡C-C≡, ≡C-O-C≡/-C(O)-O-C- (ester, ether, etc.), and ≡C-C≡ linkages in the intermediates (ME-1h/2h/3h/4h) during 5h of hydrothermal treatment at 180°C. Meanwhile, the existence of alkoxy groups such as -$OCH_3$, alkyl, -$OCH_2$-aryl, and aryl/alkyl-$COOCH_2$-aryl/alkyl groups is confirmed by the NMR peaks observed in the concentration range of δ = 3.0–4.0 ppm. Thus, this investigation highlighted the presence of ≡C-O-C≡ and -C(O)-O-C- linkages in ME, ME-1 and ME-1h/2h/3h/4h samples, which are likely used to bind aryl/alkyl units together. There was a strong spike

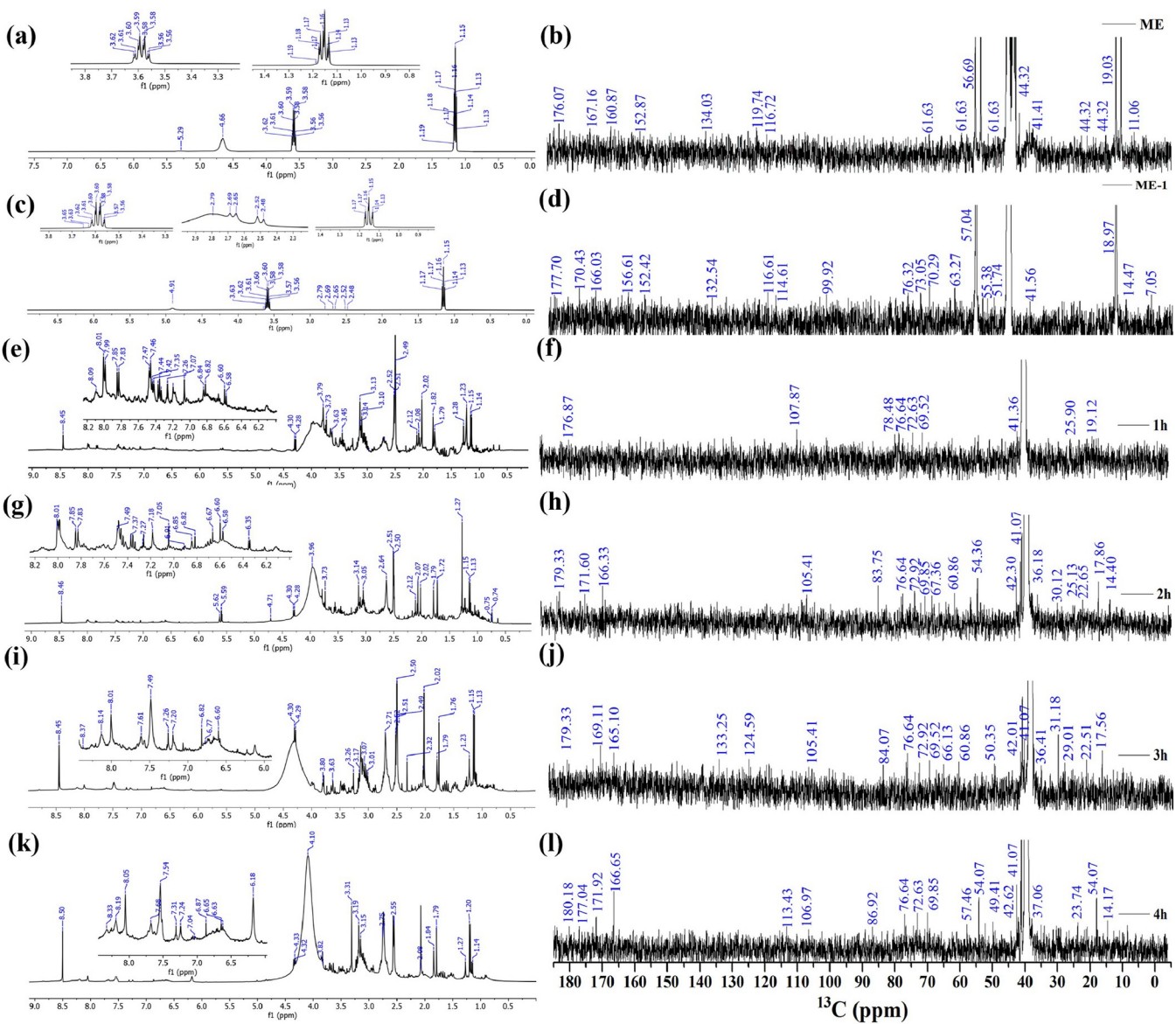

**Fig 8. $^1$H- and $^{13}$C-NMR (1D) spectra of ME, ME-1 and reaction intermediates in the time interval [1-4h].** The spectra were captured in diethyl sulfoxide-d6 solvent, and δ = 2.50 and 39.50 were taken into consideration as reference peaks.

in the range of δ = 3.0–4.0 ppm, in ME-sample derived from marigold petals and after treated ME sample for 1h, indicates the presence of alkyl (sp$^3$) content in higher concentrations as compared to ME-2h/3h/4h, respectively. Consequently, as intermediate reaction time increases (1h-4h) minimal peaks in the range δ = 3.0–4.0 ppm with reduced intensities (Fig 8C, 8E, 8G, 8I & 8K) were observed, implying that the structure of the intermediates varies at each stage.

Furthermore, the inferior intensity of these peaks in the spectrum of the N-S@MCDs sample demonstrates that the structure of N-S@MCDs is different from that of ME. Hence, the idea that the alkyl groups condensed into the aryl moieties after the hydrothermal treatment may be reasonable. Additionally, the peak associated with the alkene-type moiety connected to aryl or alkyl groups (Ar/R-CH = CH-) were detected in the range δ = 4.0–6.0 ppm in the

spectra of ME, ME-1 and ME-1h/2h/3h/4h. However, the varying intensity of these peaks, indicating the conflict for the hydrothermal pathways' capability to transform alkyl hydroxyl into alkene groups. The dehydration mechanisms used to convert alkyl alcohols to alkenes are well documented in the literature. As seen in spectrum, alkene peaks vanished from ME-4h intermediate spectrum, and similar broad-hump peak observed for ME-1h/2h/3h intermediates. These peaks emerged because of the hydroxyl groups connected to the polysaccharide-rich initial bio-precursor (i.e., marigold extract); in addition, variations in peak intensities and positions emerged as a consequence of various the hydroxyl groups' chemical surroundings. Additionally, aryl moieties' existence was established in the spectrum of ME, ME-1 and ME-1h/2h/3h/4h intermediates, by analyzing peaks between $\delta$ 6.0–8.5 ppm. Notably, the aryl peaks in the ME-4h/3h/2h/1h spectrum demonstrated greater intensity than those in the ME/ME-1 spectrum. The fact that the alkyl groups were changed into aryl moieties during the hydrothermal treatment is further supported by this. Moreover, the presence of these aryl units in the spectra of ME, ME-1 and N-S@RCD samples was confirmed by FTIR (1690 and 1590 cm$^{-1}$) and UV-Vis (240 and 345 nm) spectra. As, the UV-Vis spectra indicated a constant rise in peak intensities from 1 to 5h of reaction time (S6 Fig; see the supporting information).

Hereafter, to validate the findings from $^{1}$H-NMR spectrum, $^{13}$C-NMR spectrum of ME, ME-1 and ME-1h/2h/3h/4h likewise performed. Throughout all samples, exhibited alkyl chain (-CH$_3$, -CH$_2$, = CH-, = C =, CH$_3$CO-) peaks in the range $\delta$ = 10–50 ppm. However, the change in peak intensities demonstrated that the concentrations in ME, ME-1, and ME-1h/2h/3h/4h samples were different. In contrast to ME-1h/2h/3h/4h samples, ME displayed greater intensity peaks for -CH$_3$; -CH$_2$-; = CH-; = C =; CH$_3$CO- ($\delta$ = 10–50 ppm). Notably, it was observed that peak intensity and strength fluctuate during distinct time intervals. As the reaction time increases, the intensities and number of peaks are reduced, which is attributable to polycondensation (Fig 8D, 8F, 8H, 8J and 8I). It is conceivable due to the aromatization of alkyl groups and the synthesis of aryl groups during the hydrothermal treatment at a higher temperature (180˚C). The recognition of linkages between two carbons (Ar/C-X-C/Ar) in ME, ME-1 and ME-1h/2h/3h/4h intermediates and heteroatoms (O, N & S) was confirmed by examining peaks in the range of 50–90 ppm. The differences in peak intensities across all the samples prove that each sample has a unique alkyl and aryl structure. Additionally, by observing peaks between 100–140 ppm, the presence of sp$^2$ alkene/or aryl moieties was confirmed. And lastly, the observation of polymorphic aryl moieties is facilitated by $^{1}$H-NMR ($\delta$ = 6–8.5 ppm), FTIR (~1690 and 1590 cm$^{-1}$) and UV-Vis spectra (240 and 345 nm). Notably, the ME-4h spectrum showed aryl peaks with higher intensities than the ME, ME-1, and ME-1h/2h/3h spectrum. Furthermore, peaks corresponding to carboxyl/amide groups in the 170–180 ppm region were observed in the spectrum of all intermediates. This provides more evidence that the alkyl groups were converted into aryl moieties during the hydrothermal process.

Table 1 summarizes the characteristics of various precursor derived CQDs and compares them in terms of QY, particle size, elemental composition, and optical properties in order to correlate the features of carbon dots produced by various green biomass precursors. In conclusion, the current work reveals the transformation of temple waste (marigold extract) into N and S co-doped carbon quantum dots that display intriguing optical features, including a sizable quantum yield, sustained fluorescence, and wavelength-controlled emission. Utilizing green precursors or biomass has advantages in terms of sustainability, low cost, and mass synthesis, emerging as a potential candidate for distinct applications.

To predict the polycondensation during the hydrothermal treatment, several bulk (PXRD, elemental analysis, etc.) and molecular (UV-Vis, fluorescence, FTIR, 1D NMR, etc.) level physio-chemical characterization techniques were used in this work. The above-mentioned techniques, all supported the hypothesis that the carbonization of polysaccharide (cellulosic waste

**Table 1. Comparative table of synthesized-CQDs using various biomass precursors and their corresponding characteristics.**

| Precursor | QY (%) | Size | Elemental content (%) | | | Elemental ratio | | Optical characteristics | | Ref. |
|---|---|---|---|---|---|---|---|---|---|---|
| | | | C | N | O | N/C | O/C | Excitation wavelength (nm) | Emission wavelength (nm) | |
| Lemon juice | 28 | 4.6 | 60.9 | 15.4 | 23.7 | 0.25 | 0.38 | 533 | 631 | [42] |
| Sugarcane juice | 10.86 | 10.7 | 67 | 7.8 | 22.8 | 0.11 | 0.34 | 330 | 445 | [43] |
| Orange juice | 26 | 2.5 | - | - | - | - | 0.16 | 360 | 441 | [44] |
| Cabbage | 16.5 | 2–6 | 66.5 | 4.61 | 28.73 | 0.069 | 0.43 | 345 | 432 | [45] |
| Papaya | 18.39 | 2–6 | 74.1 | 1.7 | 23.5 | 0.02 | 0.31 | 370 | 450 | [46] |
| Peach gum | 28.46 | 2.5 | 47.6 | 14.1 | 28.6 | 0.29 | 0.6 | 370 | 445 | [47] |
| Bee pollens | 6.1–12.8 | 1.1–2.1 | 42.97 | 3.89 | 39.35 | 0.09 | 0.91 | 360 | 435 | [48] |
| Osmanthus fragrans | 18.53 | 2.23 | - | - | - | - | - | 340 | 410 | [49] |
| Rose petals | 9.6 | 4.51± 1.46 | 42.20 | 14.52 | 36.09 | 0.344 | 0.855 | 320 | 397 | [50] |
| Coffee grounds | 3.8 | 5 | - | - | - | - | - | 365 | 440 | [51] |
| Eutrophic algal blooms | 13 | 8.5±5.6 | 59 | - | 41 | - | 0.69 | 360 | 438 | [52] |
| Cow manure | 65 | 2–7 | - | - | - | - | - | 360 | 415 | [53] |
| Human pee | 5.3 | 2–4 | - | - | - | - | - | 325 | 392 | [54] |
| **Floral waste (marigold)** | **11.78** | **6.4±1.64** | **78.17** | **4.62** | **16.47** | **0.06** | **0.21** | **240, 345** | **450** | **This work** |

found in marigold extract) and other constituents present in marigold extract is the likely mechanism for the formation of CQDs from temple waste (marigold). Additionally, the hydrothermal treatment of extract components comprised hydrolysis and dehydration, which produced predominantly C5 and C6 sugars and furan (furfural, 5-hydroxymethyl furfural, etc.), respectively. Dehydration of ME was detected using the following methods: FTIR (1773 and 1695 cm$^{-1}$ for C = O and C = N, respectively); UV-Vis (240 nm and 345 nm for $\Pi$-$\Pi^*$ and n-$\Pi^*$); and the emergence of peaks and signals in 1D ($^1$H, 6.0–8.5 ppm and $^{13}$C, 100–140, 160–180 ppm) NMR. This absorbance improvement in the UV-Vis spectra of the ME-1 h to N-S@MCDs is consistent with the hydrolysis and dehydration of ME during hydrothermal treatments, which are refurbished into sugars and aromatic moieties, respectively. According to the various measuring techniques stated above, the presence of methionine aids in the hydrolysis of the acid whereas ethylenediamine aids in the dehydration of the polysaccharide isolated from marigold extract. Afterward, polymerization and polycondensation took place, as shown by a variety of bulk and molecular level analytical methods. Additionally, aromatization and carbonization resulted in the formation of aromatic clusters including nitrogen and sulphur, which was supported by NMR (1D and 2D), ATR, UV-Vis, and XPS investigations. A nuclear explosion occurred and N-S co-doped CQDs were created after the aromatic clusters reached a threshold concentration (as shown in Fig 9).

## Mechanism of latent fingerprint development (LFPs) by using N-S@MCDs/corn-starch phosphors

Powder brushing is a technique used in fingerprint analysis. It is a physical method that relies on the attraction of powder particles to the oily and moisture-laden components found in fingerprint residue, such as grease and oil. This attraction will become less powerful as more residue is cleaned up. When a fingerprint is left behind on a surface, the sweat and oil that are naturally present in the residue cause instantaneous physical adsorption of fluorescent corn-starch fingerprint powder into the ridges of the print. This allows the print to be identified and used for identification purposes. The attachment of this powder to the ridges of the fingerprint is what activates it, causing it to emit blue light. This activation is often accomplished by

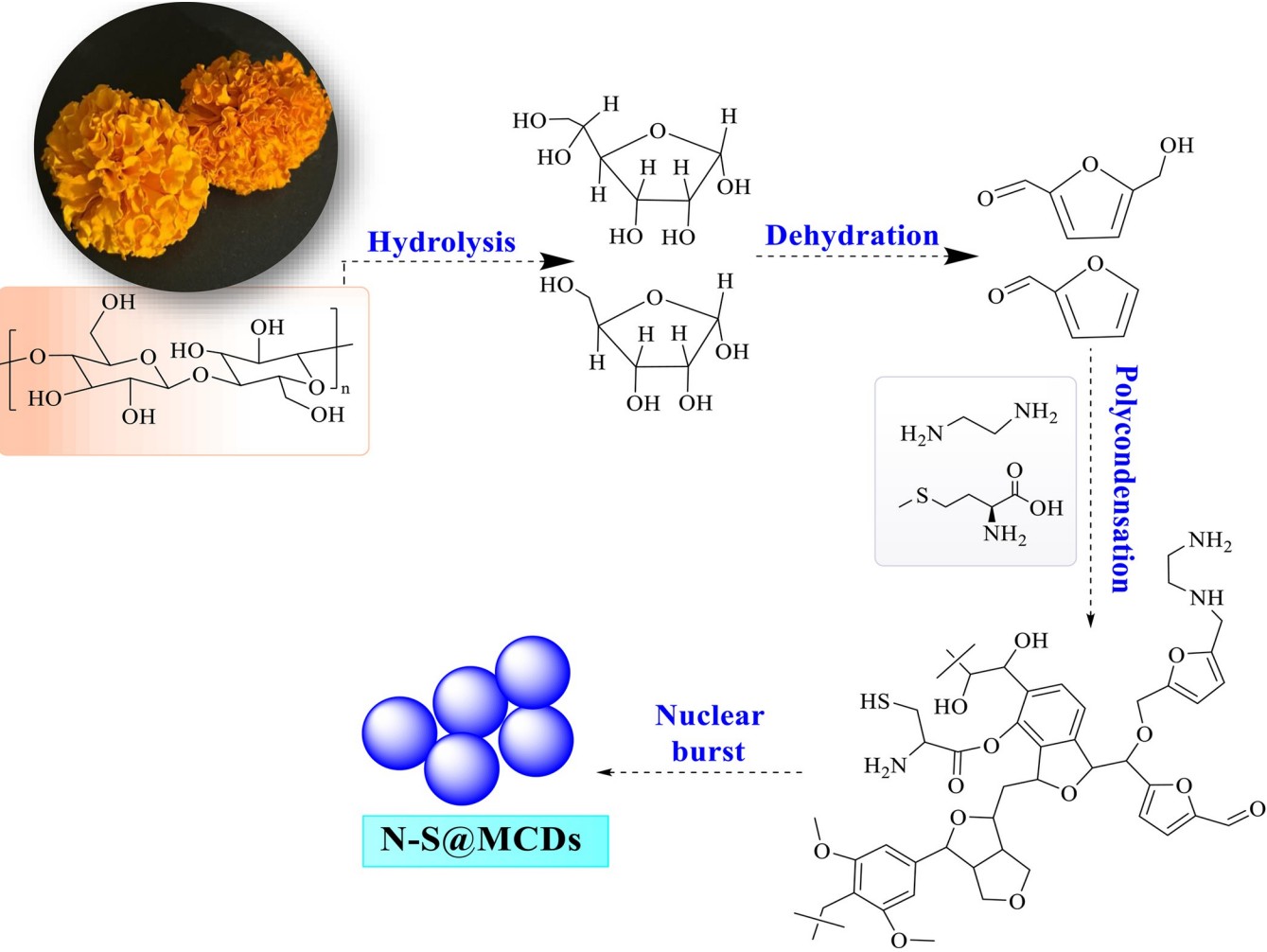

**Fig 9. Plausible synthesis mechanism of N-S@MCDs from floral waste collected from temples (marigold extract).**

subjecting the area to ultraviolet radiation having a wavelength of 365 nm. This activation makes the fingerprint more visible, providing an image that is crisp and colorful, allowing for fast study of the print. In addition, because this method allows the long-term retention of developed latent fingerprints on a wide range of surfaces, whether porous or non-porous, it is an important instrument in forensic science for the identification and examination of fingerprints.

## Visualization of developed latent fingerprints (LFPs)

In order to test the performance of N-S@MCDs-coated corn-starch for LFPs detection, five different porous and non-porous surfaces including glass, wood, plastic, metal and paper (currency), were selected. Fig 10 demonstrates that LFPs are effectively identified on all surfaces. The results evidenced that the LFPs developed by nano-carbogenic highly fluorescent fingerprint composition; N-S@MCDs/corn-starch phosphors displayed sharp, micro ridge substructure patterns without any background interference.

Fingerprint patterns can often be divided into three distinct levels. Level 1 exhibits patterns known as whorls, loops, and deltas that were not sufficiently distinguishable to be identified.

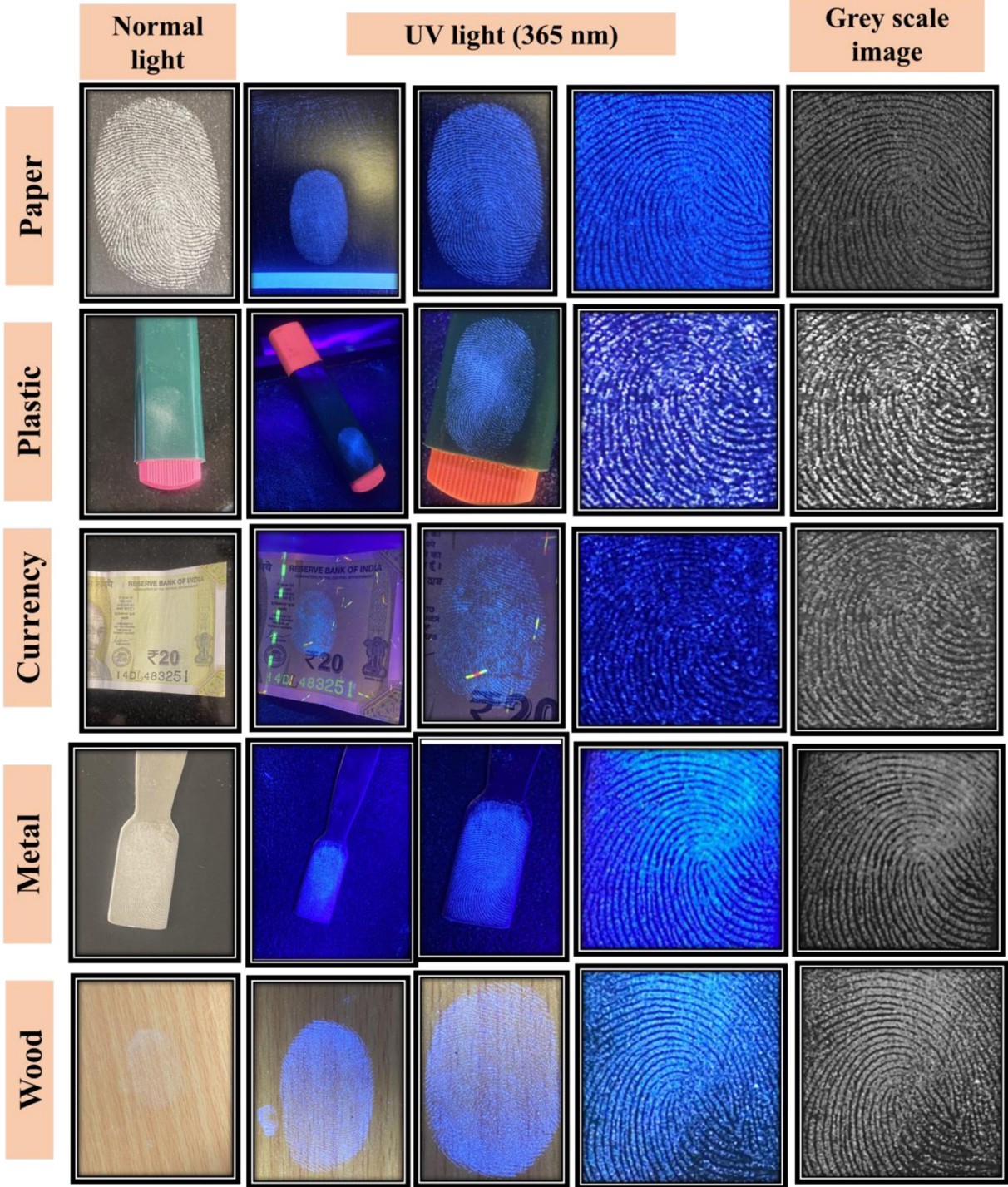

**Fig 10. LFPs developed on different porous and non-porous surfaces by using optimized N-S@MCDs/corn-starch phosphors in presence of normal light and UV light (365 nm).**

Level 2 feature points are widely recognized for their significant role in fingerprint analysis, since they assign uniqueness and invariance to fingerprints through random pairings, thereby providing vital identifying data. Level 2 characteristics minutiae points like islands, lakes,

hooks, short ridges, spurs, eyes, cores, crossovers, trifurcations, and bifurcations, which are clearly identifiable patterns. Furthermore, the feature points at level 3 frequently provide essential reference data in cases where level 2 signals are distorted or faulty, even though they cannot be solely relied upon for detecting LFPs. Level 3 kinds exhibited distinct ridges characterized by the presence of sweat pores, deviations in ridge paths, enclosure, and edge contours. These features were considered to be significant quantitative data in the process of identifying individuals [55,56]. In this study, the sensitivity of the nano-carbogenic fingerprint powder composition was assessed by observing individual latent fingerprint patterns (LFPs) using a 365 nm light source. The micro ridge substructures, consisting of three levels, were examined under UV light (365 nm) to magnify the details of various fingerprint points. These points include the delta (level 1), the core, the eye, the trifurcations, the bifurcation, the crossover, the spur, the short ridge (level 2), and the enclosure (a formation where a ridge bifurcates and re-joins in a short distance). This is illustrated in Fig 10A. The findings obtained clearly indicate that the utilization of N-S@MCDs/corn-starch phosphors for the visualization of LFPs has the potential to offer a more thorough understanding of the phenomenon, including three distinct levels. Furthermore, this approach exhibits a high degree of reliability and effectiveness in the visualization process.

Additionally, in order to examine the background interference of the optimized N-S@MCDs/corn-starch phosphors based LFPs detection, a variety of complex color porous and non-porous surfaces; paper (currency), plastic (pen highlighter), metal (scissor, spatula), wood, and glass were selected. These surfaces were used to imprint fingerprints, followed by development of LFPs with nano-carbogenic fingerprint powder composition, enabling quick visible enhancement under UV light, as well as long-term preservation of developed LFPs. Using time-correlated single photon counting (TCSPC) research, the average fluorescence lifetime of N-S@MCDs and N-S@MCDs/corn-starch phosphors was performed. S10 Fig (see supporting information) displays the N-S@MCDs and N-S@MCDs/corn-starch phosphors decay fitted by a tri-exponential function. The average fluorescence lifetime of N-S@MCDs and N-S@MCDs/corn-starch phosphors (S10 Fig.; see supporting information) was found to be $7.80 \pm 1.704$ ns and $7.51 \pm 1.463$ ns, respectively, at 350 nm excitation wavelength with an adequate $\chi^2$-value. Fig 11 shows the developed LFPs on different porous and non-porous surfaces under normal light and UV light. The ability of improved LFPs to distinguish between various ridge features without interfering from the background demonstrates how versatile, prevalent, and genuinely useful our visualization approach is for enhancing LFPs on almost all porous and non-porous surfaces.

The utilization of Python for digital processing has become an essential tool in the evaluation of latent fingerprints (LFP) images, particularly within the realm of artificial intelligence methodologies [57]. The proposed methodology relies on the complicated characteristics generated by a machine learning algorithm using the specific data points extracted from fingerprint scans. The initial LFPs image should possess sufficient distinctiveness, hence signifying the presence of unique feature structures at level 1, level 2, and level 3. Before extracting feature points, the original image of the fingerprint undergoes a three-phase transformation process to convert it into a binary model [11,12,57,58]. Firstly, a grayscale picture is produced from the input color image (Fig 12A). To accentuate the desired contour, the image is subsequently subjected to a normalization process. Finally, binaryzation is performed by classifying various pixels into "1" or "0" categories depending on their grayscale values, that is, the image's pixels are re-drown as either white (1) or black (0). Following the process of binaryzation, the regions that were once affected by noise and blurriness exhibit enhanced sharpness, while the presence of noise spots and shadows becomes imperceptible (Fig 12A).

Furthermore, Fig 12b presents additional information regarding the properties of level 2 and 3 LFPs that have been successfully retained and enhanced. These qualities comprise core,

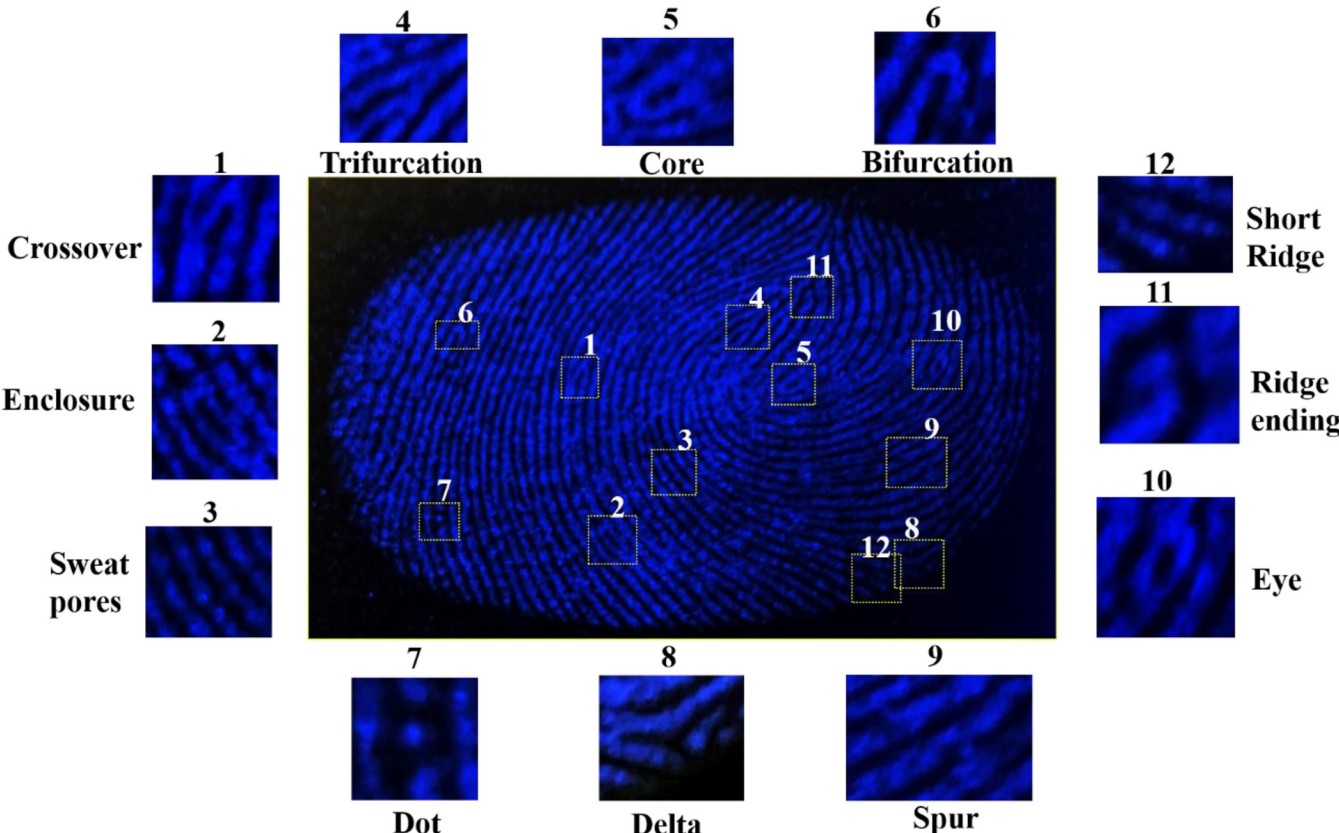

**Fig 11. Various developed micro-ridges patterns by using nano-carbogenic fingerprint powder composition; N-S@MCDs/corn-starch phosphors.**

eye, termination, bifurcation, trifurcation, and spur. In this regard, the LFP images captured on diverse porous and non-porous substrates (such as paper, metal, wood, glass, and plastic) are subjected to transformation and subsequently compared with the control image (Fig 12). In accordance with the protocols outlined by the Federal Bureau of Investigation (FBI) [26], the process of control is executed by utilizing the same finger to promptly inscribe information onto a fingerprint card using black ink. A comprehensive comprehension of the control's particulars is important for the purpose of discerning the loop, whorl, arch, or other characteristics present in fingerprints., the control is achieved by writing with the same finger immediately on a fingerprint card in black ink. In general, the control's specifics must be understood in order to identify the loop, whorl, arch, or other aspects in fingerprints. After binaryzation, the Galton-Henry Classification [59] identifies sixteen different types of 2nd-level feature points for our current data, including core (1), eye (4, 6, 7, 8, 10, 11), termination (9), trifurcation (3), bifurcation (2, 16, 5), spur (12) and extreme vital quantitative well-defined 3rd-level ridges containing sweat pores (17). In order to create a matrix for comparison, the feature point coordinates are taken from a binary picture and converted to "m × 2" values. As shown by Eq 1, Z is the sample matrix, while X is the control matrix.

$$Z = \begin{bmatrix} z_{11} \ldots z_{i1} \ldots z_{m1} \\ z_{12} \ldots z_{i2} \ldots z_{m2} \end{bmatrix}^{1/t}, \quad X = \begin{bmatrix} x_{11} \ldots x_{i1} \ldots x_{m1} \\ x_{12} \ldots x_{i2} \ldots x_{m2} \end{bmatrix}^{1/t} \tag{1}$$

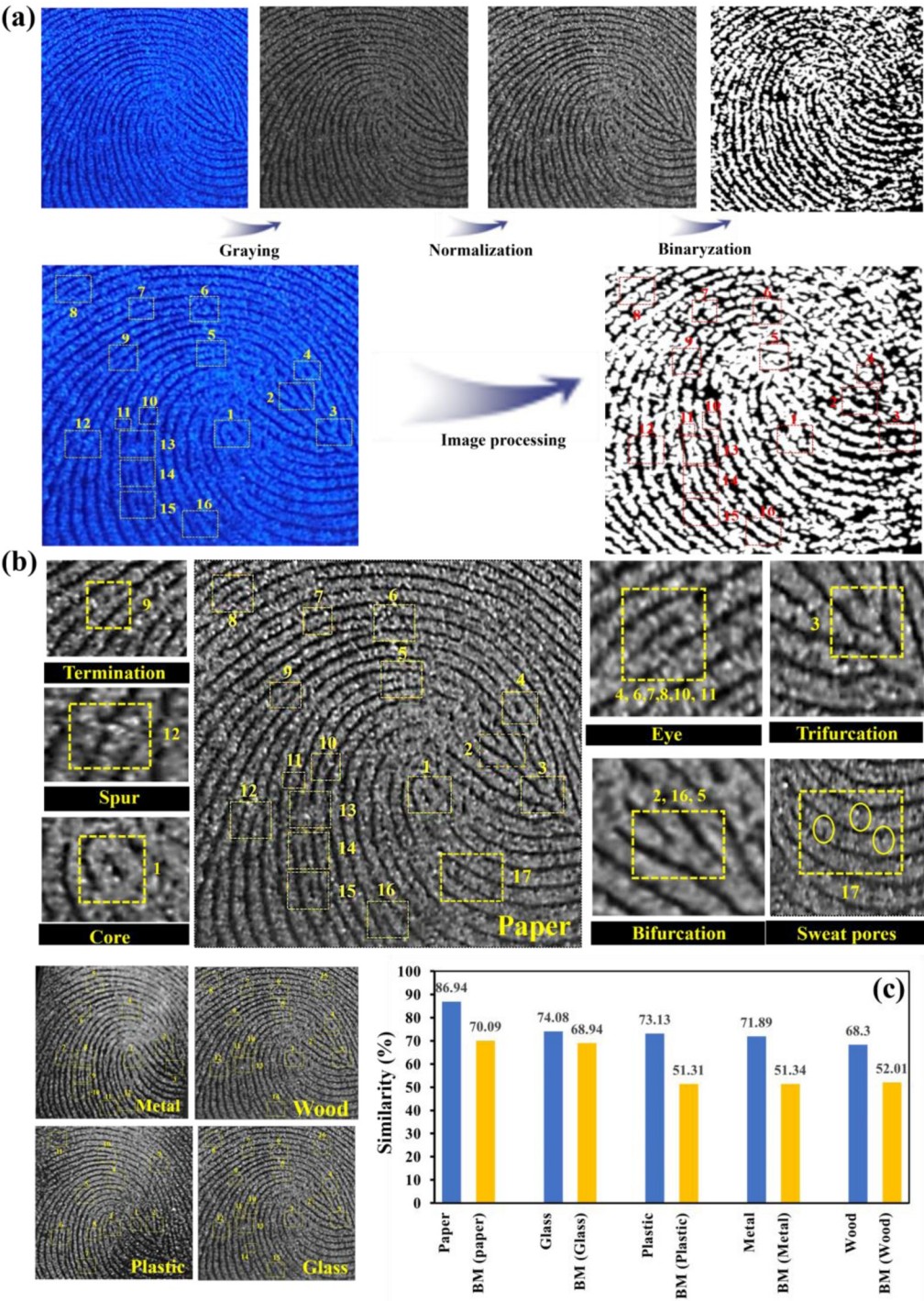

**Fig 12.** Python based image digital-processing procedure of LFPs on paper, (a) original image; grayscale image; normalized images; and binary image, (b) $2^{nd}$- $3^{rd}$ level details including 12 points (1) core, (4,6,7,8,10,11) eye, (9) termination, (12) spur, (2,16,5) bifurcation, (3) trifurcation, (17) ridges sweat pores in the partially enlarged images of LFPs on paper, and binary images of LFPs on different substrates (metal, wood, plastic, and glass) followed by similarity assessment results of same LFPs on different substrates and the benchmark obtained by dye-soaked Coomassie blue method (c).

$$P = \frac{1}{m} \sum\nolimits_{i=1}^{m} \frac{1}{\left| \frac{\sqrt{(z_{i1}-a)^2+(z_{i2}-b)^2}}{\sqrt{(x_{i1}-a)^2+(x_{i2}-b)^2}} = 1 \right| + 1}, \ (i = 1, 2, 3 \ldots, m) \qquad (2)$$

Hereafter, the Euclidean Distance Formula is used to compute the similarity of fingerprint characteristic matrices using Eq 2 to measure the similarity between the samples and the control [11,60–62]. The coordinates of the feature points in the collected values are $(z_{i1}, z_{i2})$, $(x_{i1}, x_{i2})$, and the coordinate of the image core point represented by (a, b). These two matrices match one another precisely when the matching score P is 1. The resemblance increases as the matching score increases. Based on python score (computational matching), the matching scores of our samples according to Eq (2) displays the findings as 86.94%, 71.89%, 73.13%, 74.08%, and 68.3% for LFPs on paper, metal, plastic, glass, and wood, respectively (Figs 12C and S5). As a benchmark, the classical dye-soaked cotton pads method employed for the development of LFPs on paper [28]. The matching score of this benchmark (based on Eq 2) displays the similarity index of 70.09%, 68.94%, 51.31%, 51.34% and 52.02% for LFPs on paper, metal, plastic, glass, and wood, respectively after evaluation using our digital program. As a result, the ideal pairing of N-S@MCDs/corn-starch phosphors and the artificial intelligence programme has demonstrated appreciable advancement over the conventional way for LFPs detection.

## Conclusion

Finally, the current research marks a substantial improvement in the field of forensic science and fingerprint analysis. The study revealed how *Tagetes erecta*, often known as marigold, is transformed into hetero atom-doped carbon quantum dots (N-S@MCDs), which exhibit amazing optical properties such as stable fluorescence and wavelength-tuned emission. We have uncovered the complicated process by which the components of marigold extract are aromatized and carbonized during the hydrothermal synthesis of N-S@MCDs through a series of rigorous physio-chemical analysis. In addition to this, our research has proposed a unique technique for developing latent fingerprints (LFPs) using N-S@MCDs in conjunction with corn-starch phosphors as a dusting powder. We have also created artificial intelligence software (Python) to objectively measure the degree of similarities across produced LFP images. When compared to earlier findings in fluorescence detection of LFPs, our study's distinctive capabilities become clear. We used digital image processing technologies to retain and enhance LFP feature points, resulting in a matching score that gives a robust and objective evaluation of the resemblance between the sample and benchmark images on all the porous and non-porous surfaces. An outstanding matching score of 86.94% was obtained, exceeding traditional standards and confirming the material's and approach's realistic capability for LFP detection and identification. This research not only advances our understanding of materials science, but it also provides forensic investigators with a powerful and precise instrument, opening new vistas in the field of fingerprint analysis and forensic applications.

## Supporting information

**S1 Fig. Histogram graph of N-S@MCDs, which is obtained by using 50 counts.**
(TIF)

**S2 Fig. EDAX spectrum of N-S@MCDs.**
(TIF)

**S3 Fig. EDAX spectrum of N-S@MCDs/corn-starch phosphors.**
(TIF)

**S4 Fig. Luminous starch powder under normal light and UV-light.**
(TIF)

**S5 Fig. Python cv coding screenshots of all developed fingerprints on distinct surfaces.**
(TIF)

**S6 Fig. Photographs of ME, ME-1 and N-S@MCDs under UV-light (365 nm).**
(TIF)

**S7 Fig.** LC-MS chromatogram of ME (a), ME-1 (b & c) and N-S@MCDs (d &e).
(TIF)

**S8 Fig. Relative PL intensity of N-S@MCDs/corn-starch phosphors after 1 day of synthesis and after 6 months.**
(TIF)

**S9 Fig. Photographs of N-S@MCDs/corn-starch phosphors after 1 day of synthesis and after 6 months of synthesis.**
(TIF)

**S10 Fig. Time-resolved photoluminescence decay of N-S@MCDs (sample 1) and N-S@MCDs/corn-starch phosphors (sample 2) ($\lambda_{ex}$ = 350 nm and $\lambda_{em}$ = 450 nm) in aqueous medium with an adequate fitting.**
(TIF)

**S1 Table. Elemental analysis of N-S@MCDs and N-S@MCDs/corn-starch phosphors.**
(DOCX)

**S2 Table. Green precursor-derived CQDs and their related quantum yield (QY).**
(DOCX)

**S1 Graphical abstract.**
(PDF)

## Acknowledgments

The authors duly acknowledge Amity University Uttar Pradesh, Noida, India and Dr. Anil Kumar Mishra, Director, Institute of Nuclear Medicine and Allied Sciences, Defence Research and Development Organization of India, New Delhi, India for his time-to-time guidance and providing the instrumentation facilities as external co-guide of Ms. Nisha Yadav.

## Author Contributions

**Conceptualization:** Nisha Yadav, Vivek Mishra.

**Data curation:** Nisha Yadav, Amarnath Mishra, Vivek Mishra.

**Formal analysis:** Nisha Yadav, Deeksha Mudgal, Amarnath Mishra, Sacheendra Shukla, Vivek Mishra.

**Investigation:** Nisha Yadav, Vivek Mishra.

**Methodology:** Nisha Yadav, Vivek Mishra.

**Project administration:** Tabarak Malik, Vivek Mishra.

**Resources:** Nisha Yadav, Vivek Mishra.

**Software:** Nisha Yadav, Vivek Mishra.

**Supervision:** Vivek Mishra.

**Writing – original draft:** Nisha Yadav, Vivek Mishra.

**Writing – review & editing:** Vivek Mishra.

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
