## [Decision Letter · Decision Letter 0]

13 Sep 2023

PONE-D-23-18687Fluorescent Carbon Quantum Dots Produced from Natural Resources for Sweat Latent Fingerprint Recognition Using Machine Learning AlgorithmsPLOS ONE

Dear Dr. Malik,

Thank you for submitting your manuscript to PLOS ONE. After careful consideration, we feel that it has merit but does not fully meet PLOS ONE’s publication criteria as it currently stands. Therefore, we invite you to submit a revised version of the manuscript that addresses the points raised during the review process.

We look forward to receiving your revised manuscript.

Kind regards,

Sivasankar Koppala

Academic Editor

PLOS ONE

2. Please note that PLOS ONE has specific guidelines on code sharing for submissions in which author-generated code underpins the findings in the manuscript. In these cases, all author-generated code must be made available without restrictions upon publication of the work. Please review our guidelines at https://journals.plos.org/plosone/s/materials-and-software-sharing#loc-sharing-code and ensure that your code is shared in a way that follows best practice and facilitates reproducibility and reuse."

4. We note that Figures 1, 3, 9, 10, 11, 12, S4, S6 and S9 in your submission contain copyrighted images. All PLOS content is published under the Creative Commons Attribution License (CC BY 4.0), which means that the manuscript, images, and Supporting Information files will be freely available online, and any third party is permitted to access, download, copy, distribute, and use these materials in any way, even commercially, with proper attribution. For more information, see our copyright guidelines: http://journals.plos.org/plosone/s/licenses-and-copyright.

1. You may seek permission from the original copyright holder of Figures 1, 3, 9, 10, 11, 12, S4, S6 and S9 to publish the content specifically under the CC BY 4.0 license.

Reviewers' comments:

Reviewer's Responses to Questions

**Comments to the Author**

1. Is the manuscript technically sound, and do the data support the conclusions?

Reviewer #1: Yes

Reviewer #2: Yes

Reviewer #3: Partly

2. Has the statistical analysis been performed appropriately and rigorously? 

Reviewer #1: N/A

Reviewer #2: Yes

Reviewer #3: Yes

3. Have the authors made all data underlying the findings in their manuscript fully available?

Reviewer #1: Yes

Reviewer #2: Yes

Reviewer #3: Yes

4. Is the manuscript presented in an intelligible fashion and written in standard English?

Reviewer #1: Yes

Reviewer #2: Yes

Reviewer #3: Yes

5. Review Comments to the Author

Reviewer #1: The authors utilized biomass to prepare the carbon dots for fingerprint applications. The experimental work was well-planned and written systematically. The manuscript can be accepted after doing the following corrections.

1. Please refine the language carefully

2. Please check the section numbers.

3. Align the table 1 properly.

4. Improve the quality of all figures.

Reviewer #2: In this work，marigold was chosen as the synthetic precursor of carbon dots, and elucidated the conversion mechanism of marigold extract to N-S@MCDs by analyzing the composition of intermediates formed. And use digital processing tools to accurately detect LFP. However, there are still some questions and suggestions：

1、Why is the carbon dot in Figure 1b a region, not a single particle? In Figure 1f, why is it not magnified after compounding to clearly show that the carbon dots are on the starch surface?

2、The value range of the abscissa in Figure 6 remains the same, from large to small or small to large.

3、A large part of the article is about the formation and recombination process of carbon dots. Is there any innovation in the mechanism part?

4、As for the matching score of LFP, what is the current level of technology? 86.94% is just the value of one substrate, and the effect of other substrates is not enough to reach this level. Could it be used in practice?

Reviewer #3: Introduction

1. Driving inspiration from nature to take design cues for the preparation of functional materials, to address the grave issues facing the sustainable production of nanomaterials without employing harmful substances, is imperative to use an environmentally benign and renewable raw material source. Carbon quantum dots (CQDs) are a greener example of a functionalized material produced from processing biomass with a distinctive natural composition with fascinating features. The field of naturally occurring CQDs has expanded tremendously; nevertheless, due to inadequate characterization of green precursors, flexible and heterogeneous in nature, the synthesis methodology, and repeatability remain a bottleneck. Utilizing edible green resources for the synthesis of CQDs, such as tomato, banana, milk, orange, cabbage, corn bran, and coconut, has recently attracted a lot of attention[

The sentence formation is not proper and neither linked properly. It is written poorly. Choices of words are inadequate.

What is green resources? It should be replaced with adequate word. This word has been repeated so many times.

2. The majority of CQDs formed from biomass are blue-emitting.

Incomplete sentence.

3. To obtain good mobility and avoid dye powder stagnation and substrate adhesion, these dyes are indeed dispersed in a significant volume of media (powder carrier) in practical applications. Based on the coffee ring effect and the unquenched CQDs fluorescence during the drying process, we also revealed a spraying technique that induces LFPs on impregnable surfaces by utilizing CQDs.

4. In this work, we developed a nano-carbogenic fluorescent fingerprint powder composition (NS@MCDs/corn-starch phosphors) for developing LFPs. Such powder comprises environment friendly highly fluorescent carbon quantum dots (N-S@MCDs) synthesized from biomass (marigold extract). These N-S@MCDs, when incorporated on the corn starch powder using a greener approach, enables the development of highly fluorescent fingerprint patterns on various non-porous surfaces, their quick intensification under UV light to display a brilliant, sharp fingerprint, and their long-term preservation. We used a digital-processing tool to accurately detect LFPs, which retrieved the fine characteristics of LFPs on various substrates, a precise comparison of the characteristics with the control, and then displayed the corresponding scores on a computer. The highest possible matched score was 86.94%. Moreover, we elucidate the conversion mechanism of marigold extract to N-S@MCDs by analyzing the composition of intermediates formed. Translating these promising research findings into real-world technologies with substantial social and economic relevance, necessitates a collaborative, openminded approach involving materials scientists, biologists, and forensic investigators.

2. Experimental Section

Synthesis of hetero-atom doped carbon quantum dots (N-S@MCDs)

5. After being lyophilized obtained sticky black product, the resulting supernatant (N-S@MCDs) was appropriately redispersed in water.

Sentence formation

2.2.2. Preparation of N-S@MCDs/corn-starch phosphors

6. N-S@MCDs and corn-starch are mixed in a mortar followed by the addition of Milli Q water as a binder by ultrasonic agitation over 10 min at RT

Check Grammar

2.3. Characterization and Instrumentations

7. Morphology. The high-resolution X-ray diffraction (XRD) pattern was produced using a Cu K radiation source and a PANalytical Empyrean X-ray diffractometer at an acceleratingpotential of 40 kV, in the 2θ range of 10-60° with a Cu Kα radiation source. X-rays were acquired of solid samples that had been finely powdered. Transmission electron microscopy (TEM) and high-resolution transmission electron microscopy were used to assess the surface morphology and particle size of N-S@MCDs (HRTEM). TEM images were obtained on a

JEOL JEM-2100F electron microscope with a 200 kV acceleration voltage. The material was

drop-cast onto a copper grid that had been coated with carbon, and then the samples were

allowed to dry at room temperature. Scanning Electron Microscope (SEM) with EDAX

(Energy-dispersive X-ray) spectroscopy was done from the University of Delhi using Jeol JSM 6610LV using Tungsten electron sources at 10KV with Magnification X5 to X 3,00,000 with High Vacuum mode using LN2 free EDS detector. 1H and 13C NMR (400 MHz for proton and 100 MHz for carbon) spectra were recorded with a JEOL spectrometer using trimethylsilane as an internal standard.

8. The associated d-spacing of N-S@MCDs is higher than the bulk graphite i.e., 3.34 Å, which depicts turbostratic carbon structure (sp2 deformation). Additionally, one more peak centered around 42.50° (2θ) represents a predominantly graphitic structure with an interlayer spacing of 0.21 nm[26]. Moreover, TEM and HR-TEM images were utilized to examine particle shape, size, and nature of N-S@RCD. Fig. 3[a-f] shows that as-prepared N-S@MCDs were uniform, mono-dispersed spheres with a mean diameter of 6.4±1.64 nm

Page 14- variation in font

9. The particle size distribution of assynthesized N-S@MCDs is shown in Fig. S1, based on 50 counts. The HR-TEM image validates the N-spherical S@RCD's form, and it also shows some particles to have well resolved lattice fringes (0.35 nm, d-spacing).

Where is fig. S1??

10. Additionally, the EDAX spectrum confirms the existence of C, N, O and S in N-S@MCDs and NS@ MCDs/corn-starch phosphors (Fig. S2 and Fig. S3, respectively). Elemental analysis of NS@ MCDs and N-S@MCDs-coated corn-starch was carried out and the results were consolidated in Table S1.

(Page 17)

11. Typically, there are three levels in which fingerprint patterns can be classified. Level 1 displays whorl, loop and delta which were not distinct enough to recognized. However, the level two feature points, which are well-known for providing the most crucial identifying data in fingerprint analysis by assigning uniqueness and invariance to fingerprints through random combinations. Level 2 features refer to minute points like islands, lakes, hooks, short ridges, spur, eye, core, crossover, trifurcations, and bifurcations that were obviously recognizable patterns.

Explain level 1 and whether level 2 is minutiae points or minute point??

12. The fact that the improved LFPs clearly show different ridge details without background interference shows how adaptable, ubiquitous, and realistically helpful our visualization technique is for improving LFPs on nearly all porous and non-porous surfaces.

Page 25- rewrite

13. Python digital processing has grown to be a crucial tool for evaluating LFPs pictures as one of the artificial intelligence approaches [54]. This approach is based on the minute characteristics that are created by a machine learning programme from the local information points inherent in fingerprint scans. The original LFPs image should be distinct enough, which indicates that it has distinct level 1 and level 2 feature structures. The fingerprint's original picture is transformed into a binary model in three phases prior to extracting feature points [8,9,54,55]. Firstly, a grayscale picture is created from the input color image (Fig. 12). To emphasize the target contour, the picture is then normalized. Finally, binaryzation is accomplished by classifying various pixels into "1" or "0" categories based on their grayscale values, that is, the image's pixels are redrew as either white (1) or black (0). After binaryzation, the noisy (blur) areas get sharper while the noise spots and shadows vanish (Fig. 12). Additionally, Fig. 12 provides further information about the level 2 and 3 LFPs.

Not explained properly.

grammar

level 3 characteristics are?

6. PLOS authors have the option to publish the peer review history of their article (what does this mean?). If published, this will include your full peer review and any attached files.

Reviewer #1: No

Reviewer #2: No

Reviewer #3: No

---

## [Author Response · Author response to Decision Letter 0]

27 Oct 2023

PLOS ONE’s- 

Journal Requirements-

1. Please ensure that your manuscript meets PLOS ONE's style requirements, including those for file naming. The PLOS ONE style templates can be found at https://journals.plos.org/plosone/s/file?id=wjVg/PLOSOne_formatting_sample_main_body.pdf andhttps://journals.plos.org/plosone/s/file?id=ba62/PLOSOne_formatting_sample_title_authors_affiliations.pdf

Response: We have arranged the manuscript as per PLOS ONE’s style requirements, including file name.

2. Please note that PLOS ONE has specific guidelines on code sharing for submissions in which author-generated code underpins the findings in the manuscript. In these cases, all author-generated code must be made available without restrictions upon publication of the work. Please review our guidelines at https://journals.plos.org/plosone/s/materials-and-software-sharing#loc-sharing-code and ensure that your code is shared in a way that follows best practice and facilitates reproducibility and reuse."

Response: As per suggestion, we have followed the guidelines.

Response: Repository information should be reflected at acceptance. I have changed it during resubmission. The data generated or used during the study are available from the corresponding author by request.

4. We note that Figures 1, 3, 9, 10, 11, 12, S4, S6 and S9 in your submission contain copyrighted images. All PLOS content is published under the Creative Commons Attribution License (CC BY 4.0), which means that the manuscript, images, and Supporting Information files will be freely available online, and any third party is permitted to access, download, copy, distribute, and use these materials in any way, even commercially, with proper attribution. For more information, see our copyright guidelines: http://journals.plos.org/plosone/s/licenses-and-copyright.

Response: Now, there is no copyright image (Figures 1, 3, 9, 10, 11, 12, S4, S6 and S9) in our manuscript, it is either captured by our own digital camera or SEM, TEM images of our own compounds. Fingerprints in figures 10, 11, and 12 are the main author’s fingerprint so there is no need to take copyright permission.

1. You may seek permission from the original copyright holder of Figures 1, 3, 9, 10, 11, 12, S4, S6 and S9 to publish the content specifically under the CC BY 4.0 license.

Response: Now, there is no copyright image (Figures 1, 3, 9, 10, 11, 12, S4, S6 and S9) in our manuscript, it is either captured by our own digital camera or SEM, TEM images of our own compounds. Fingerprints in figures 10, 11, and 12 are the main author’s fingerprint so there is no need to take copyright permission.

Response: Now, there is no copyright image in our manuscript.

Reviewer comments- 

Reviewer #1: 

The authors utilized biomass to prepare the carbon dots for fingerprint applications. The experimental work was well-planned and written systematically. The manuscript can be accepted after doing the following corrections.

1. Please refine the language carefully

Response: As per the suggestions, we have carefully refined the language.

2. Please check the section numbers.

Response: We have arranged it as per PLOS ONE format.

3. Align the table 1 properly.

Response: As per suggestions, we have arranged Table 1 properly.

4. Improve the quality of all figures. 

Response: As per suggestions, we are providing high resolution images around 600 DPI.

Reviewer #2: 

In this work, marigold was chosen as the synthetic precursor of carbon dots and elucidated the conversion mechanism of marigold extract to N-S@MCDs by analyzing the composition of intermediates formed. And use digital processing tools to accurately detect LFP. However, there are still some questions and suggestions：

1、Why is the carbon dot in Figure 1b a region, not a single particle? In Figure 1f, why is it not magnified after compounding to clearly show that the carbon dots are on the starch surface?

Response: Carbon dots structures is just a prediction based on the identification of the functional group present which is tested by XPS analysis and FTIR spectrum as well as SEM-EDAX data. Due to its high agglomeration effect, we cannot say that it can be freely present as a single particle, it can always be in a group of particles. Secondly, starch is a porous material and after integration of carbon dots onto the surface of corn starch which is formulated from the corn kernel were found to be oval in shape and have smooth surfaces, ruptured due to hydroxyl bonding of synthesized carbon dots onto the surface and in the voids of corn-starch which is also confirmed by their SEM images. This explanation is already incorporated in the manuscript text.

2、The value range of the abscissa in Figure 6 remains the same, from large to small or small to large.

Response: Thank you for highlighting it, we have arranged the value range of the abscissa from small to large.

3、A large part of the article is about the formation and recombination process of carbon dots. Is there any innovation in the mechanism part?

Response: Our innovative part is the easiest way to develop a fingerprint powder using bioresources. Yes, there is the innovative part where we have tried to prove the plausible mechanism in detail by developing carbon dots using the extract of marigold flower at different time interval (1-4h) using NMR spectroscopy and LC-MS chromatogram to investigate its precursor and its intermediate behaviour. 

4、As for the matching score of LFP, what is the current level of technology? 86.94% is just the value of one substrate, and the effect of other substrates is not enough to reach this level. Could it be used in practice?

Response: Assessing the "current state of technology" for matching Latent Fingerprints (LFPs) is a matter influenced by considerable variability attributable to the diverse techniques and equipment employed across different law enforcement agencies and forensic science laboratories. In the specific domain of comparing fingerprint samples, elevated matching scores are indicative of a heightened degree of resemblance between the samples. These matching scores, a common tool for such assessments, are crafted to gauge the extent of similarity.

The 86.94% value we report in our study, particularly for the paper surface, surpasses the benchmark score (i.e., 70.09%) established by the dye soaked Coomassie blue method, which presently serves as the standard in use by several law enforcement agencies and forensic science laboratories. However, it is noteworthy that other surfaces have also exhibited favourable results, with the exception being the rough wood surface.

As per suggestion, we have repeated the similarity index of developed fingerprint on various surfaces and compared against the benchmark too using Coomassie blue (dye) utilized in classical dye-soaked cotton pads method by various law enforcement agencies and forensic laboratories. On all the surfaces our developed fluorescent powder is showing better similarity index percentage (Figure 12 c). So, definitely, it can be used in the practice, and we would also like to highlight that it is filed for the Indian Patent and will be transfer to the industry for the commercialization of this N-S@MCD as fluorescent powder (developing LFPs powder).

Reviewer #3: 

Introduction

1. Driving inspiration from nature to take design cues for the preparation of functional materials, to address the grave issues facing the sustainable production of nanomaterials without employing harmful substances, is imperative to use an environmentally benign and renewable raw material source. Carbon quantum dots (CQDs) are a greener example of a functionalized material produced from processing biomass with a distinctive natural composition with fascinating features. The field of naturally occurring CQDs has expanded tremendously; nevertheless, due to inadequate characterization of green precursors, flexible and heterogeneous in nature, the synthesis methodology, and repeatability remain a bottleneck. Utilizing edible green resources for the synthesis of CQDs, such as tomato, banana, milk, orange, cabbage, corn bran, and coconut, has recently attracted a lot of attention.

The sentence formation is not proper and neither linked properly. It is written poorly. Choices of words are inadequate.

What are green resources? It should be replaced with adequate word. This word has been repeated so many times.

Response: We have tried to make it properly and tried to make a story by linking with each other. Removed the “Green resources” with more adequate words.

2. The majority of CQDs formed from biomass are blue emitting. Incomplete sentence.

Response: We have tried to make it complete and modified whole sentence.

3. To obtain good mobility and avoid dye powder stagnation and substrate adhesion, these dyes are indeed dispersed in a significant volume of media (powder carrier) in practical applications. Based on the coffee ring effect and the unquenched CQDs fluorescence during the drying process, we also revealed a spraying technique that induces LFPs on impregnable surfaces by utilizing CQDs.

Response: We have improved it as per suggestions.

4. In this work, we developed a nano-carbogenic fluorescent fingerprint powder composition (NS@MCDs/corn-starch phosphors) for developing LFPs. Such powder comprises environment friendly highly fluorescent carbon quantum dots (N-S@MCDs) synthesized from biomass (marigold extract). These N-S@MCDs, when incorporated on the corn starch powder using a greener approach, enables the development of highly fluorescent fingerprint patterns on various non-porous surfaces, their quick intensification under UV light to display a brilliant, sharp fingerprint, and their long-term preservation. We used a digital-processing tool to accurately detect LFPs, which retrieved the fine characteristics of LFPs on various substrates, a precise comparison of the characteristics with the control, and then displayed the corresponding scores on a computer. The highest possible matched score was 86.94%. Moreover, we elucidate the conversion mechanism of marigold extract to N-S@MCDs by analyzing the composition of intermediates formed. Translating these promising research findings into real-world technologies with substantial social and economic relevance, necessitates a collaborative, openminded approach involving materials scientists, biologists, and forensic investigators.

Response: We have improved it as per suggestions.

2. Experimental Section

Synthesis of hetero-atom doped carbon quantum dots (N-S@MCDs)

Response: We have improved it as per suggestions.

5. After being lyophilized obtained sticky black product, the resulting supernatant (N-S@MCDs) was appropriately redispersed in water. Sentence formation

Response: We have improved it as per suggestions.

2.2.2. Preparation of N-S@MCDs/corn-starch phosphors

6. N-S@MCDs and corn-starch are mixed in a mortar followed by the addition of Milli Q water as a binder by ultrasonic agitation over 10 min at RT. Check Grammar

Response: We have improved it as per suggestions.

2.3. Characterization and Instrumentations

7. Morphology. The high-resolution X-ray diffraction (XRD) pattern was produced using a Cu K radiation source and a PANalytical Empyrean X-ray diffractometer at an accelerating potential of 40 kV, in the 2θ range of 10-60° with a Cu Kα radiation source. X-rays were acquired of solid samples that had been finely powdered. Transmission electron microscopy (TEM) and high-resolution transmission electron microscopy were used to assess the surface morphology and particle size of N-S@MCDs (HRTEM). TEM images were obtained on a JEOL JEM-2100F electron microscope with a 200 kV acceleration voltage. The material was

drop-cast onto a copper grid that had been coated with carbon, and then the samples were

allowed to dry at room temperature. Scanning Electron Microscope (SEM) with EDAX

(Energy-dispersive X-ray) spectroscopy was done from the University of Delhi using Jeol JSM 6610LV using Tungsten electron sources at 10KV with Magnification X5 to X 3,00,000 with High Vacuum mode using LN2 free EDS detector. 1H and 13C NMR (400 MHz for proton and 100 MHz for carbon) spectra were recorded with a JEOL spectrometer using trimethylsilane as an internal standard.

Response: We have improved it as per suggestions.

8. The associated d-spacing of N-S@MCDs is higher than the bulk graphite i.e., 3.34 Å, which depicts turbostratic carbon structure (sp2 deformation). Additionally, one more peak centered around 42.50° (2θ) represents a predominantly graphitic structure with an interlayer spacing of 0.21 nm [26]. Moreover, TEM and HR-TEM images were utilized to examine particle shape, size, and nature of N-S@RCD. Fig. 3[a-f] shows that as-prepared N-S@MCDs were uniform, mono-dispersed spheres with a mean diameter of 6.4±1.64 nm. Page 14- variation in font.

Response: We have improved it as per suggestions.

9. The particle size distribution of as-synthesized N-S@MCDs is shown in Fig. S1, based on 50 counts. The HR-TEM image validates the N-spherical S@RCD's form, and it also shows some particles to have well resolved lattice fringes (0.35 nm, d-spacing). Where is fig. S1??

Response: We have modified it and give complete information about Fig. S1 location.

10. Additionally, the EDAX spectrum confirms the existence of C, N, O and S in N-S@MCDs and NS@ MCDs/corn-starch phosphors (Fig. S2 and Fig. S3, respectively). Elemental analysis of NS@ MCDs and N-S@MCDs-coated corn-starch was carried out and the results were consolidated in Table S1. (Page 17)

Response: We have improved it as per suggestions and mentioned the location of figures and table.

11. Typically, there are three levels in which fingerprint patterns can be classified. Level 1 displays whorl, loop and delta which were not distinct enough to recognized. However, the level two feature points, which are well-known for providing the most crucial identifying data in fingerprint analysis by assigning uniqueness and invariance to fingerprints through random combinations. Level 2 features refer to minute points like islands, lakes, hooks, short ridges, spur, eye, core, crossover, trifurcations, and bifurcations that were obviously recognizable patterns. Explain level 1 and whether level 2 is minutiae points or minute point??

Response: We have improved it as per suggestions and explained it in detail.

12. The fact that the improved LFPs clearly show different ridge details without background interference shows how adaptable, ubiquitous, and realistically helpful our visualization technique is for improving LFPs on nearly all porous and non-porous surfaces. Page 25- rewrite

Response: We have improved it as per suggestions.

13. Python digital processing has grown to be a crucial tool for evaluating LFPs pictures as one of the artificial intelligence approaches [54]. This approach is based on the minute characteristics that are created by a machine learning programme from the local information points inherent in fingerprint scans. The original LFPs image should be distinct enough, which indicates that it has distinct level 1 and level 2 feature structures. The fingerprint's original picture is transformed into a binary model in three phases prior to extracting feature points [8,9,54,55]. Firstly, a grayscale picture is created from the input color image (Fig. 12). To emphasize the target contour, the picture is then normalized. Finally, binaryzation is accomplished by classifying various pixels into "1" or "0" categories based on their grayscale values, that is, the image's pixels are redrew as either white (1) or black (0). After binaryzation, the noisy (blur) areas get sharper while the noise spots and shadows vanish (Fig. 12). Additionally, Fig. 12 provides further information about the level 2 and 3 LFPs. Not explained properly. grammar level 3 characteristics are?

Response: We have improved it and highlighted as per suggestions.

---

## [Decision Letter · Decision Letter 1]

10 Dec 2023

Harnessing Fluorescent Carbon Quantum Dots from Natural Resource for Advancing Sweat Latent Fingerprint Recognition with Machine Learning Algorithms for Enhanced Human Identification

PONE-D-23-18687R1

Dear Dr. Malik,

We’re pleased to inform you that your manuscript has been judged scientifically suitable for publication and will be formally accepted for publication once it meets all outstanding technical requirements.

Kind regards,

Sivasankar Koppala

Academic Editor

PLOS ONE

Additional Editor Comments (optional):

Reviewers' comments:

Reviewer's Responses to Questions

**Comments to the Author**

1. If the authors have adequately addressed your comments raised in a previous round of review and you feel that this manuscript is now acceptable for publication, you may indicate that here to bypass the “Comments to the Author” section, enter your conflict of interest statement in the “Confidential to Editor” section, and submit your "Accept" recommendation.

Reviewer #1: All comments have been addressed

Reviewer #2: All comments have been addressed

2. Is the manuscript technically sound, and do the data support the conclusions?

Reviewer #1: Yes

Reviewer #2: Yes

3. Has the statistical analysis been performed appropriately and rigorously? 

Reviewer #1: N/A

Reviewer #2: Yes

4. Have the authors made all data underlying the findings in their manuscript fully available?

Reviewer #1: Yes

Reviewer #2: Yes

5. Is the manuscript presented in an intelligible fashion and written in standard English?

Reviewer #1: Yes

Reviewer #2: Yes

6. Review Comments to the Author

Reviewer #1: The authors have addressed all the comments and improved the manuscript. The manuscript can be accepted in its present form.

Reviewer #2: (No Response)

7. PLOS authors have the option to publish the peer review history of their article (what does this mean?). If published, this will include your full peer review and any attached files.

Reviewer #1: No

Reviewer #2: No

---

## [Editor Report · Acceptance letter]

26 Dec 2023

PONE-D-23-18687R1 

PLOS ONE

Dear Dr. Malik, 

I'm pleased to inform you that your manuscript has been deemed suitable for publication in PLOS ONE. Congratulations! Your manuscript is now being handed over to our production team.

Kind regards, 

on behalf of

Dr. Sivasankar Koppala 

Academic Editor

PLOS ONE